# Antibiotic-induced acceleration of type 1 diabetes alters maturation of innate intestinal immunity

Xue-Song Zhang[1,2]*, Jackie Li[1,2], Kimberly A Krautkramer[3], Michelle Badri[1,2,4], Thomas Battaglia[1,2], Timothy C Borbet[1,2], Hyunwook Koh[5], Sandy Ng[1,2], Rachel A Sibley[1,2], Yuanyuan Li[6], Wimal Pathmasiri[6], Shawn Jindal[1,2], Robin R Shields-Cutler[7], Ben Hillmann[7], Gabriel A Al-Ghalith[7], Victoria E Ruiz[1,2], Alexandra Livanos[1,2], Angélique B van 't Wout[8], Nabeetha Nagalingam[8], Arlin B Rogers[9], Susan Jenkins Sumner[6], Dan Knights[7], John M Denu[3], Huilin Li[5], Kelly V Ruggles[1,2], Richard Bonneau[4], R Anthony Williamson[8], Marcus Rauch[8], Martin J Blaser[1,2,10]*

[1]Department of Medicine, New York University Langone Medical Center, New York, United States; [2]Human Microbiome Program, New York University Langone Medical Center, New York, United States; [3]Department of Biomolecular Chemistry, Wisconsin Institute for Discovery, University of Wisconsin School of Medicine and Public Health, Madison, United States; [4]Center for Data Science, New York University, New York, United States; [5]Department of Population Health, New York University Langone Medical Center, New York, United States; [6]Nutrition Research Institute, University of North Carolina at Chapel Hill School of Public Health, Kannapolis, United States; [7]Computer Science and Engineering, BioTechnology Institute, University of Minnesota, St. Paul, United States; [8]Janssen Prevention Center London, Janssen Pharmaceutical Companies of Johnson and Johnson, London, United Kingdom; [9]Department of Biomedical Sciences, Cummings School of Veterinary Medicine, Tufts University, North Grafton, United States; [10]Department of Microbiology, New York Uniersity Langone Medical Center, New York, United States

*For correspondence:
Xuesong.Zhang@nyumc.org (X-SZ);
Martin.Blaser@nyumc.org (MJB)

**Competing interests:** The authors declare that no competing interests exist.

**Abstract** The early-life intestinal microbiota plays a key role in shaping host immune system development. We found that a single early-life antibiotic course (1PAT) accelerated type 1 diabetes (T1D) development in male NOD mice. The single course had deep and persistent effects on the intestinal microbiome, leading to altered cecal, hepatic, and serum metabolites. The exposure elicited sex-specific effects on chromatin states in the ileum and liver and perturbed ileal gene expression, altering normal maturational patterns. The global signature changes included specific genes controlling both innate and adaptive immunity. Microbiome analysis revealed four taxa each that potentially protect against or accelerate T1D onset, that were linked in a network model to specific differences in ileal gene expression. This simplified animal model reveals multiple potential pathways to understand pathogenesis by which early-life gut microbiome perturbations alter a global suite of intestinal responses, contributing to the accelerated and enhanced T1D development.
DOI: https://doi.org/10.7554/eLife.37816.001

**eLife digest** The human body contains many microbes that play important roles in our health. These microbes begin to live in the intestines, skin, and mouth shortly after birth. They form complex communities called the microbiome, which changes as babies develop. The microbiome works with organs to maintain human health. For example, the lower intestinal tract is home to the most numerous and active microbes in the body. The intestines provide microbes with food and a welcoming environment, and the microbes make products the body needs, influence immune system development, and help maintain a balance of beneficial microbes.

Use of antibiotics to treat infections, particularly early in life, disrupts intestinal microbe communities. Recent studies show that such microbiome disturbances may affect how the immune system develops and the rate at which type 1 diabetes develops. Type 1 diabetes is an autoimmune disease in which the immune system destroys cells in the pancreas that produce insulin. Scientists would like to learn more about how use of antibiotics in early life may contribute to the development of this disease.

Now, Zhang et al. show that a single course of antibiotics administered early in life accelerates the development of type 1 diabetes in mice prone to develop the disease. In the experiments, a strain of laboratory mice that spontaneously develops type 1 diabetes were either given a single course of antibiotics, three courses of antibiotics, or no antibiotics in their first weeks of life. After one single course, the gut microbiome was different in mice treated with antibiotics compared with mice who were never exposed. The antibiotics also changed the molecules produced by these microbes. These alterations in the microbiome turned on or off certain genes in the intestine, affecting the development of the immune system.

Zhang et al. identified some microbes that appear to protect against type 1 diabetes and others that seem to speed it up and how they do so. Antibiotic use in children is very common, so finding ways to reduce its potentially harmful effects on development are critical. The experiments provide one way to study how antibiotics may contribute to autoimmune disease. It also may allow scientists to test ways to reverse harmful change.

DOI: https://doi.org/10.7554/eLife.37816.002

## Introduction

The human gastrointestinal (GI) tract contains a microbiome of enormous cell number (*Sender et al., 2016*) and complexity (*Lozupone et al., 2012*), which plays important roles in shaping development of host immunity (*Honda and Littman, 2016*; *Hooper et al., 2012*; *Kinnebrew and Pamer, 2012*). Altered composition of the GI microbiota modifies risk of inflammatory conditions—including type 1 diabetes (T1D), asthma, and inflammatory bowel disease—by perturbing immune system development (*Eberl et al., 2015*; *Flak et al., 2013*; *Fujimura and Lynch, 2015*; *Gensollen et al., 2016*; *Kostic et al., 2015*; *Kriegel et al., 2011*; *Livanos et al., 2016*; *Rooks and Garrett, 2016*; *Schulfer et al., 2018*).

T1D is characterized by T-cell-mediated destruction of pancreatic β-cell-containing islets (*Wilson et al., 1998*), but the triggers and intermediary molecular mechanisms remain unclear. Since numbers of intestinal Treg cells are significantly reduced in T1D, altered gut microbiota might play an initiating role (*Badami et al., 2011*). The worldwide increasing incidence of T1D, with decreasing age of onset (*Patterson et al., 2012*; *Paun et al., 2017*; *Wändell and Carlsson, 2013*), coincides with the widespread use of antibiotics in children (*Hersh et al., 2011*; *Lee et al., 2014*). Since antibiotic exposure affects the intestinal microbiota, potentially changing interplay with immune systems, it could contribute to the rise in T1D; recent studies in the NOD (non-obese diabetic) mouse model support this hypothesis (*Brown et al., 2016*; *Candon et al., 2015*; *Hu et al., 2017*; *Livanos et al., 2016*). In male NOD mice, three courses of a pulsed (macrolide) antibiotic treatment (3PAT) altered the intestinal microbiota and reduced intestinal lamina propria Th17- and Treg-populations, accelerating T1D development (*Livanos et al., 2016*). That isolated cecal contents from the antibiotic-exposed NOD mice transferred to germ-free recipient mice produced parallel immunological effects, further supports a causal role of the antibiotic-perturbed microbiota in T1D pathogenesis (*Livanos et al., 2016*). Enhanced T1D induction depended on the antibiotics used (*Brown et al.,*

*2016*; *Candon et al., 2015*; *Hu et al., 2017*), suggesting that differences in their activities influenced overall effects. Although the roles of antibiotics perturbing the microbiome and promoting T1D are becoming defined, the underlying molecular mechanisms require better resolution.

A single early-life PAT course altered the intestinal microbiota and specific intestinal T-cell populations and effectors in C57BL/6 mice; experiments involving germ-free mice showed that the perturbed microbiota was both necessary and sufficient for the effects (*Ruiz et al., 2017*). Here we asked whether the same single early-life (pup day of life P5-P10) antibiotic pulse was sufficient to enhance T1D in NOD mice. This work now shows that the extensive early-life effects of the brief antibiotic course on the microbiota initiate a global cascade of effects flowing from the gut lumen via metabolites and specific interactions with host cells that change the developmental program of innate and adaptive immunity, leading to accelerated and enhanced T1D.

## Results

### A single early-life antibiotic exposure accelerated T1D development

To evaluate the effects of antibiotic exposure on T1D incidence, NOD mice were given a single (1PAT) or three courses (3PAT) of a macrolide antibiotic, or not (controls) (*Figure 1A*). In both control groups, females had higher T1D incidence than males, as expected (*Bao et al., 2002*; *Livanos et al., 2016*; *Markle et al., 2013*). In males, T1D development in the two control groups was similar, but both antibiotic-exposed groups had significantly accelerated and enhanced T1D rates (*Figure 1B*); their similarity indicates the sufficiency of the first exposure for the full effect. In the female controls, the spontaneous T1D rates approached 80%, and neither antibiotic exposure significantly increased the rates. As such, we focused on male mice especially the 1PAT group in whom the exposure ended by P10. That the median T1D development was at P147 provided a prolonged window to understand the intermediary mechanisms. Examining the pancreatic islets, at P42, there were no significant differences between the PAT and control groups, but by P70, significantly more islets showed inflammation in both PAT groups than controls (*Figure 1C*), confirming that the enhanced pathological process was well-advanced by P70.

1PAT exposures significantly reduced body weight for both male and female mice continuing to P70 (*Figure 1—figure supplement 1*), and similar to 3PAT exposure (data not shown), but not at later time points. There was no relationship between early-life body weight and risk or timing of T1D development.

### Single early-life PAT persistently altered the intestinal microbiome

We then examined the antibiotic effects on the intestinal microbiome at points prior to the observed insulitis. On P12, two days after 1PAT ended, there were significant changes in the cecal and fecal microbiota persisting until at least P49 (*Figure 2* and *Figure 2—figure supplement 1*). Community structure (β-diversity) markedly differed between controls and both PAT groups (*Figure 2A* and *Figure 2—figure supplement 1B*); further studies focused on the simpler 1PAT experiment. In all cecal and fecal samples tested, 1PAT suppressed α-diversity weeks after the exposure ended (*Figure 2B* and *Figure 2—figure supplement 1A,C*). Thus, the single early life pulse led to a persistent change in the microbial community.

1PAT also increased the inter-subject microbial heterogeneity. Although the composition of the gut microbiomes of control males and females were nearly identical, there were significant differences in microbiome heterogeneity amongst the 1PAT-exposed mice (*Figure 2—figure supplements 2* and *3*). In the ileum, PAT and control differed less from each other than in the cecum (*Figure 2—figure supplement 4*), and there were no significant differences between males and females for each treatment (data not shown).

### 3PAT changes fecal microbiome community structure and richness

The 3PAT exposure accelerated T1D development, which is consistent with our previous observations (*Livanos et al., 2016*). Also consistent with the prior observations, we saw significant changes in fecal microbiota after course 1 (at P21), course 2 (at P35), and course 3 (at P49). β-diversity was markedly different between control and 3PAT, and α-diversity was significantly decreased at all three early timepoints (*Figure 2—figure supplement 5*). Whereas the control males and females were

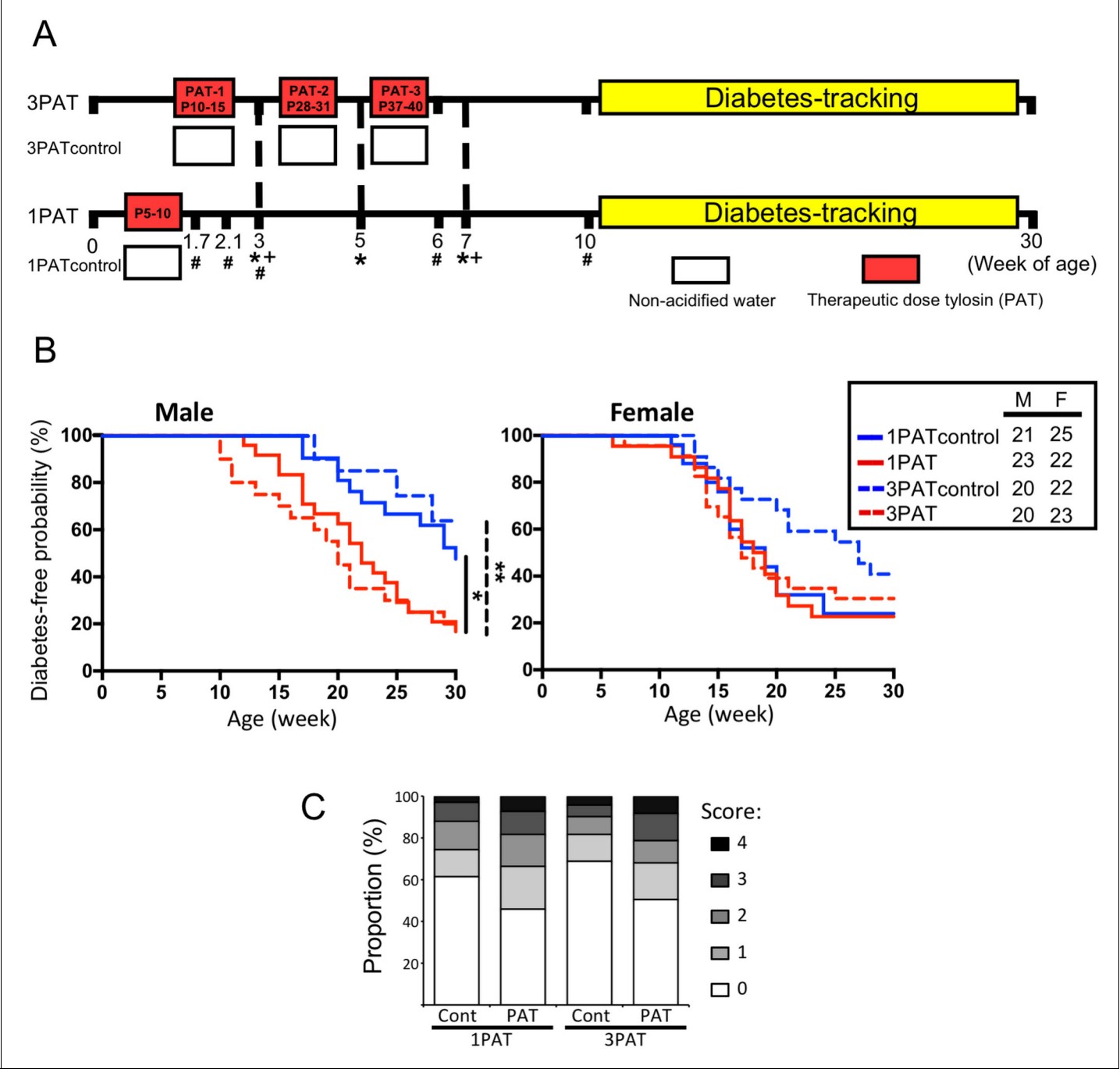

**Figure 1.** Effect of 1PAT/3PAT on T1D development. (**A**) Design of NOD 1PAT and 3PAT experiments. Pregnant NOD/ShiLtJ mice were randomized into four groups: 3PAT, 1PAT, and their controls (3PAT control, 1PAT control) as described in the Materials and methods section. From 11–30 weeks of age, mice were tested weekly for T1D by blood glucose measurement. *Weeks at which fecal samples microbiome were analyzed by16S rRNA sequencing; and + metagenomic sequencing. #Timepoint at which a subset of mice were sacrificed for collection of ileal and cecal tissues and content. Microbiome was analyzed by 16S rRNA sequencing and metagenomics, and tissues for gene expression, metabolomics, and flow cytometric analysis. (**B**) Kaplan-Meier analysis of T1D incidence in male and female NOD mice. Statistical significance was determined by the log-rank test. *p=0.019; **p=0.002. (**C**) Insulitis in male NOD mice at P70. Upon necropsy, the pancreas was preserved and stained, as described in Materials and methods, and insulitis determined with scores 0–4 indicating progressive infiltration (*Livanos et al., 2016*). The extent of pancreatic islet inflammation (scores 1–4) in PAT is higher than in controls at P70 in both the 1PAT and 3PAT experiments, but differences by mouse (n = 6/group) were not significant. Score differences were significant across the 235 individual 1PAT, and 229 control islets, (p=0.0016, Chi square), and across 201 individual 3PAT and 210 control islets (p=0.0034). [See also *Figure 1—figure supplement 1*].

DOI: https://doi.org/10.7554/eLife.37816.003

*Figure 1 continued on next page*

*Figure 1 continued*

The following source data and figure supplement are available for figure 1:

**Source data 1.** Time points of T1D development and scores of pancreatic islets.
DOI: https://doi.org/10.7554/eLife.37816.005
**Figure supplement 1.** Effect of 1PAT on NOD mouse body weight.
DOI: https://doi.org/10.7554/eLife.37816.004

nearly identical in inter-subject heterogeneity, there were significant differences in the 3PAT-exposed mice at all three timepoints. There were no significant differences in α-diversity between males and females for either the control or 3PAT mice (*Figure 2—figure supplement 6*).

Comparing across the 1PAT and 3PAT experiments based on Shannon index analysis, α-diversities were similar in the two control groups, but were significantly lower in the 1PAT than the 3PAT group in both males (*Figure 2—figure supplement 7*), and in females (data not shown).

## Effects of PAT on specific taxa

With the observed inter-subject heterogeneity (*Figure 2A* and *Figure 2—figure supplement 5A*), we next asked which specific taxa were associated with accelerated T1D. In total, across the 518 fecal samples at the three early time points studied, we identified 76 individual taxa. Using the stringent ANCOM algorithm (*Mandal et al., 2015*), findings in males and females were similar, as were comparisons of the 1PAT and 3PAT mice with their controls (*Figure 2C*); thus the major effects on taxa were conserved across the treatments and both sexes were affected similarly (*Figure 2—figure supplements 3* and *6*). These results pointed to a differential transduction of the effects of the altered microbiome in the male and female hosts (see below). In essentially all of the 12 individual timepoint/group comparisons, the relative abundances of four taxonomic groups (*Enterococcus*, *Blautia*, Enterobacteriaceae, and *Akkermansia*) were significantly over-represented in PAT. In contrast, the relative abundances of four taxa (S24-7, Clostridiales, *Oscillospira*, and *Ruminococcus*) were significantly under-represented in ≥ 7 time-point/treatment comparisons between PAT and control. No other representational differences were reproducible across treatment and time, nor were significant differences identified between males and females (*Figure 2—figure supplements 3* and *6*). Analysis of the mixed effect of time in different groups further confirmed the over-represented taxa (*Enterococcus*, *Blautia*, Enterobacteriaceae, and *Akkermansia*) and the under-represented taxa (S24-7 and *Ruminococcus*) in PAT, and revealed another under-represented taxon *Anaeroplasma*. This analysis revealed representational differences prior to the phenotypic events in a small group of taxa strongly linked with accelerated T1D or with protection in the male mice.

## PAT directionally alters the metagenome and its metabolic products

To determine whether the PAT-altered microbiota differed in metabolic functions or whether the taxonomic differences merely led to functional substitutions, we examined the fecal metagenome in 24 1PAT and control mice at both P21 and P49. Based on Bray-Curtis analysis of 300 metabolic pathways identified in the shotgun sequencing results, 1PAT significantly affected metagenomic composition (*Figure 2D*). Notably, of the 131 pathways differentiating 1PAT and control mice, 97% were overrepresented in the 1PAT samples, significantly deviated from chance at both times in both males and females (p<0.001 for each subanalysis) (*Figure 2—figure supplements 8* and *9*).

In unsupervised hierarchical clustering of the pathways with the highest inter-individual variance, we identified a cluster strongly enriched for 1PAT (20/25 samples), that contains 34 specific pathways (*Figure 2—figure supplement 8*), encoding genes involved in long chain carbohydrate degradation, specific amino acid biosynthesis, and bacterial cell structural components (e.g. peptidoglycan synthesis and maturation) (*Figure 2—figure supplement 8*). In males, of 50 pathways that differed significantly between 1PAT and control, 22 (44%) were significant at both P21 and P49. For females, of 51 significant pathways, only 9 (18%) were shared (*Figure 2—figure supplement 9* and *Supplementary file 1*). Of 49 pathways significant at P21, 10 (20%) were shared between males and females, but at P49, 19 (35%) of 54 were shared. At P21, of 39 pathways that were significantly differential in males, 100% were greater in PAT than controls. In females, 20 were differential, but 95% were greater in PAT than controls. Thus, there was a marked asymmetry in the pathways in

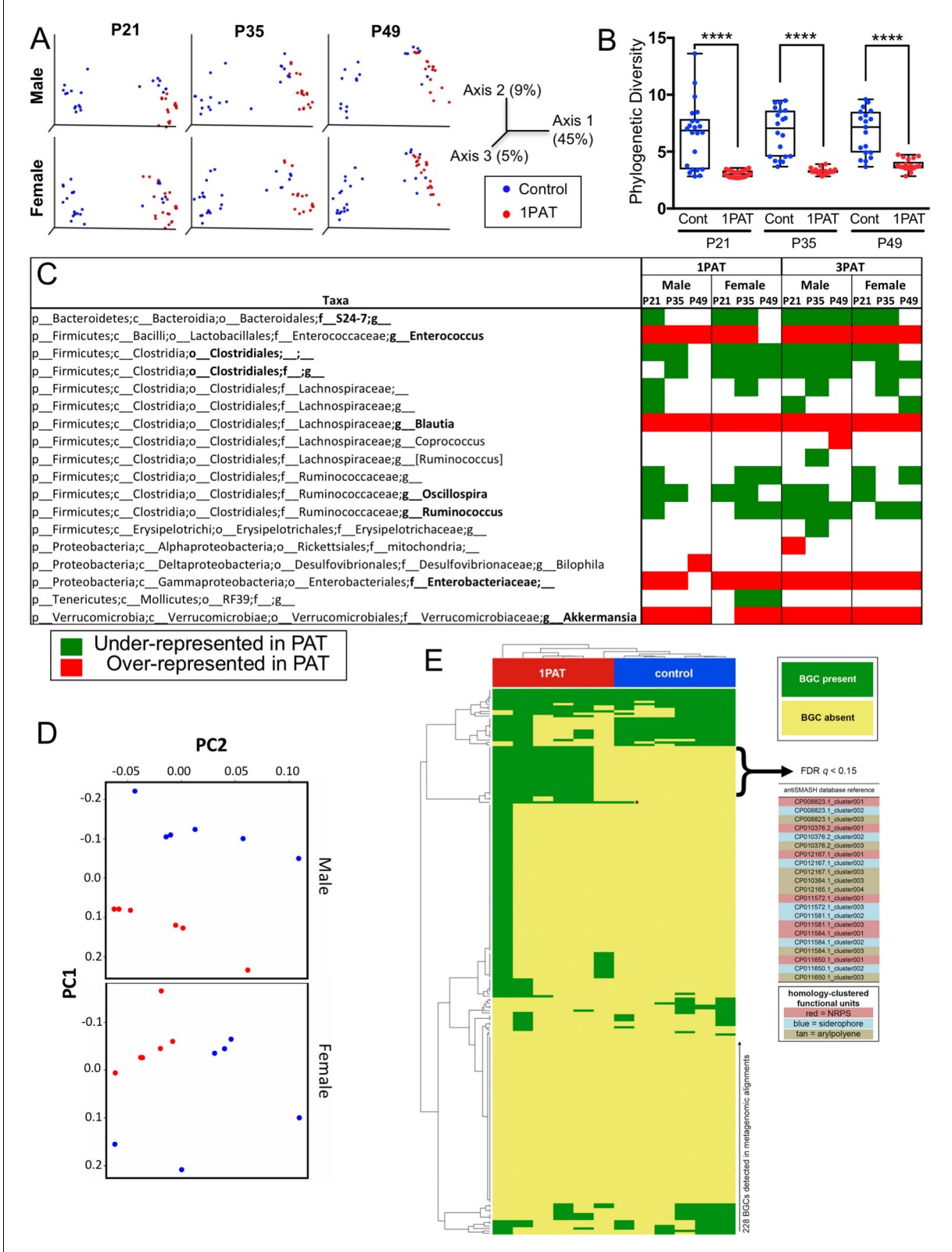

**Figure 2.** Early-life fecal microbiota in PAT and control NOD mice. (**A**) β-diversity, as determined by unweighted UniFrac analysis of control and 1PAT fecal microbiota over time, analyzed by QIIME2. Inter- and intra-group UniFrac distances were all significant (p<0.0001), determined by one-way-ANOVA with Tukey correction for multiple comparisons. (**B**) α-diversity (Phylogenic diversity (PD)) of fecal microbiota in control and 1PAT males. Statistical significance determined by one-way-ANOVA test. ****p<0.0001. (**C**) Early-life taxa significantly under- or over-represented in 1PAT and 3PAT

*Figure 2 continued on next page*

*Figure 2 continued*

mice compared to controls, by ANCOM in QIIME2. Fecal specimens were obtained at P21, P35, and P49 from male and female mice, exposed or not to 1PAT or 3PAT (n = 20–25 mice per group). (D) Metagenomic (MetaCyc) pathway distribution in male and female mice at P21, visualized by principal component analysis (PCA; blue, control; red, 1PAT, n = 6 samples per group). (E) Comparison of BGC content in male 1PAT and control metagenomic P21 samples. Presence (blue) or absence (gold) heat map of the 228 detected BGCs in male 1PAT (green) vs. control (purple) mice, generated by using an accelerated optimal gapped alignment algorithm against a BGC database (*Al-Ghalith and Knights, 2017*; *Blin et al., 2017*; *Needleman and Wunsch, 1970*). For visualization and dendrograms, axes are clustered by average linkage. For each of the 228 BGCs shown, a Fisher's exact test was performed on a contingency table of 1PAT/control vs. BGC presence/absence. Filtering to an FDR q < 0.15 yielded 22 uncharacterized BGCs enriched in 1PAT vs. control mice, and an *Enterococcus* polysaccharide pathway, indicated by '*'. Similarity-based clustering using custom Python and C code collapses the 22 uncharacterized BGCs to three functional groups (indicated in table by color) annotated to the family Enterobacteriaceae, encoding putative aryl polyene, NRPS (nonribosomal peptide synthetase), and siderophore pathways. [See also *Figure 2—figure supplements 1–11*, and *Supplementary file 1*].

DOI: https://doi.org/10.7554/eLife.37816.006

The following source data and figure supplements are available for figure 2:

**Source data 1.** Values of alpha diversities and beta diversities.

DOI: https://doi.org/10.7554/eLife.37816.018

**Figure supplement 1.** Fecal and cecal microbiota characteristics of 1PAT and control male and female NOD mice.

DOI: https://doi.org/10.7554/eLife.37816.007

**Figure supplement 2.** Cecal microbiota characteristics of male and female control and PAT NOD mice.

DOI: https://doi.org/10.7554/eLife.37816.008

**Figure supplement 3.** Fecal microbiota characteristics of male and female NOD mice in 1PAT and control groups in early life.

DOI: https://doi.org/10.7554/eLife.37816.009

**Figure supplement 4.** Ileal microbiota characteristics of male and female control and PAT NOD mice.

DOI: https://doi.org/10.7554/eLife.37816.010

**Figure supplement 5.** Microbiota characteristics of 3PAT and control NOD mice in early life.

DOI: https://doi.org/10.7554/eLife.37816.011

**Figure supplement 6.** Microbiota characteristics of male and female NOD mice in 3PAT and control groups in early life.

DOI: https://doi.org/10.7554/eLife.37816.012

**Figure supplement 7.** α-diversity in males across the 1PAT and 3PAT experiments at P21-P49.

DOI: https://doi.org/10.7554/eLife.37816.013

**Figure supplement 8.** Metagenomic analysis of microbiota characteristics of 1PAT and control NOD mice in early life.

DOI: https://doi.org/10.7554/eLife.37816.014

**Figure supplement 9.** Metagenomic analysis of microbial pathways in fecal samples differing by treatment and by sex.

DOI: https://doi.org/10.7554/eLife.37816.015

**Figure supplement 10.** Heat maps of the correlation of the representation of fecal metagenomic pathways at P21.

DOI: https://doi.org/10.7554/eLife.37816.016

**Figure supplement 11.** Biosynthetic gene cluster (BGC) frequencies in 1PAT and control samples at P21 and P49.

DOI: https://doi.org/10.7554/eLife.37816.017

both males and females, with overrepresentation in PAT; 10 (53%) of the 19 over-represented pathways in females also were over-represented in males. Those 10 pathways were related to bacterial biosynthesis of the amino acids lysine, methionine, and homoserine and related to the bacterial degradation of rhamnose, aspartate, and inositol. Lysine provides one basis for arginine metabolism, which plays important roles in immune regulation (*Peranzoni et al., 2007*); plasma lysine is substantially produced by intestinal microbes (*Metges et al., 1999*). Homoserine is an essential part of the Gram-negative enteric bacterial quorum-sensing auto-inducer, homoserine lactone, which mediates communication between bacteria and could have immunodulatory roles (*Gaida et al., 2016*). As such, up-regulation of these intestinal bacterial amino acid pathways by 1PAT could affect intestinal and systemic homeostasis in the pups, and may affect subsequent T1D onset.

An alternative way to consider these pathways is to assess how closely the representation for one matches the others. Pairwise correlations between 14 fecal metagenomics pathways were significantly different at P21 (*Figure 2—figure supplement 10*), identifying a close and significant correlation between superpathways of sulfate assimilation and cysteine biosynthesis, phospholipid biosynthesis, purine nucleotide salvage, oleate biosynthesis, (saturated) fatty acid elongation, and hexitol fermentation in all male samples at P21 (*Figure 2—figure supplement 10A*). The remaining eight metagenomic pathways showed no significant associations. The majority of these relationships

remained when considering 1PAT males at P21 independent of controls, but did not reach significance, likely due to the reduced sample size (*Figure 2—figure supplement 10B,C*). These relationships were not seen in the control-only analysis.

Since secondary metabolites are bioactive small molecules affecting microbial community structure and/or host physiology (*Dorrestein et al., 2014*; *Sharon et al., 2014*), we then asked whether the metagenomic analysis could also identify biosynthetic gene clusters (BGCs) encoding significantly differential secondary metabolites. Using an accelerated optimal gapped alignment algorithm, we mapped the metagenomic reads against a BGC database and identified 228 BGCs with high rates of within-sample metagenomic coverage. The number of BGCs per sample varied widely (*Figure 2—figure supplement 11*), and alone did not significantly distinguish 1PAT in male or female mice (p>0.05, Mann Whitney U test). Then asking whether the presence of particular BGCs were distinguishing at the earliest time point, P21, we found 23 BGCs significantly enriched in 1PAT male mice (*Figure 2E*). One enriched BGC mapped to a polysaccharide product in the MIBiG database, and is produced by *Enterococcus* species (*Medema et al., 2015*; *Xu et al., 1998*). Clustering the other 22 uncharacterized BGCs by sequence homology (*Rashidi et al., 2018*; *Shields-Cutler et al., 2018b*), collapsed them to three functional units predicted to produce a triscatecholate siderophore biosynthesis pathway member, a siderophore secondary metabolite, and an arylpolyene, all annotated to the family *Enterobacteriaceae*. In total, these studies indicate a directional (not substitutional) effect of PAT on the metabolite profiles as detected by metagenomic analyses, and are consistent with changes in taxa that were independently identified in the analysis of 16S relative abundances.

To determine whether the altered metagenome affects important microbial products, we examined production of seven short chain fatty acids (SCFA) in cecal samples at P23 and P42. At P23, the 1PAT mice had significantly reduced levels of butyric and propionic acid (p<0.05 for both) compared to controls (*Figure 3A*); none of the other SCFAs were significantly different. By P42, there no longer were significant differences in any of the tested SCFA (data not shown). Thus, antibiotic exposure, by altering the taxonomic and metagenomic composition, reduced two important host-signaling microbial metabolites in early life.

## PAT affects host metabolism

We next asked whether the 1PAT-altered microbiome affected host metabolic phenotypes. Using samples obtained from P15 to P42, we identified 30 metabolites that were significantly different between 1PAT and control in serum, and 12 in liver (*Figure 3B* and *Supplementary file 2*). These included three metabolites (uracil, citric acid, and isoleucine) at significantly higher levels in both serum and liver, and a fourth (valine) that was less abundant in hepatic samples from both the P23 and P42 PAT-exposed mice. The altered amino acid levels, consistent with the metagenomic representation of amino acid biosynthetic pathways (*Supplementary file 1*), provide direct evidence that the PAT-altered microbiome produces a metabolic signal that is transduced into the host.

## Developmental differences in intestinal gene expression by sex

Given that 1PAT altered microbial populations and taxa, microbial genes, and metabolites, we next asked how 1PAT affects the microbial interaction with host tissues that led to accelerated T1D. We focused on the immunologically active ileum since our prior 3PAT studies showed altered gene expression in the NOD mouse ileum, and that immunological effects could be transferred to recipient mice only using the perturbed microbiome (*Livanos et al., 2016*). Further, experiments in C57BL/6 mice confirmed that a 1PAT-altered microbiome is both necessary and sufficient for such immunological changes (*Ruiz et al., 2017*).

We began by characterizing ileal gene expression in P2 pups, asking how males and females differ in earliest life, prior to antibiotic exposure. Bray-Curtis analysis of RNA-Seq data indicated that ileal gene expression profiles are distinct at a global level between male and female P2 mice, and significantly differed from P12 (*Figure 4A* and *Figure 4—figure supplement 1*). At P2, we identified the ~1500 genes that comprised the significant sex-specific differences (*Figure 4B*), including KEGG pathways involved in the mTOR, MAPK, B cell receptor, and T cell receptor signaling. These sex-specific differences at post-natal day 2, preceding any antibiotic exposure, may provide an important

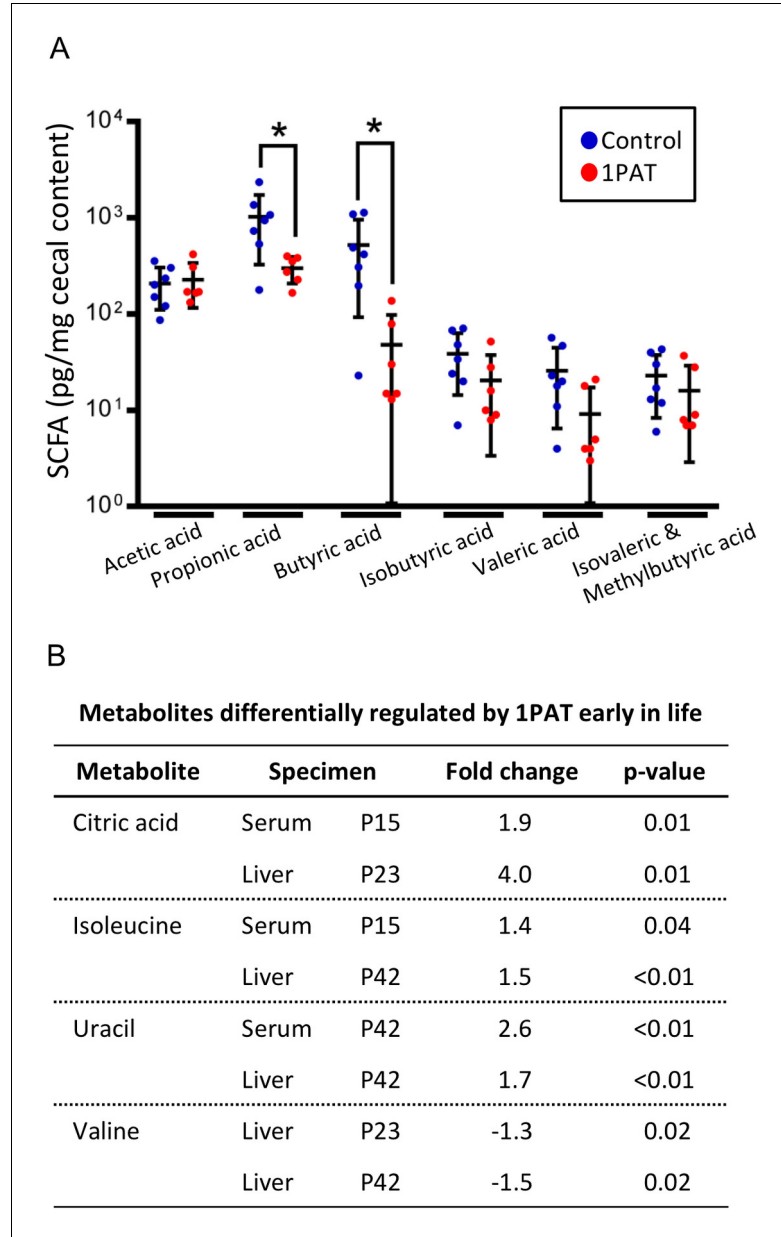

**Figure 3.** Effect of 1PAT on early-life metabolism. (**A**) Quantitation of six short chain fatty acids (SCFA) in P23 cecal samples. Samples were examined by targeted GC/MS; formic acid was not detectable (not shown). Groups: Control (n = 7), 1PAT(n = 6), compared by Welch's T Test for unpaired samples (*p<0.05). (**B**) Hepatic and serum metabolites differentially induced by 1PAT. Analysis based on significance at $\geq$ 2 time points or in both serum and liver in the same direction (n = 7–17 samples per group) [See also *Supplementary file 2*].
DOI: https://doi.org/10.7554/eLife.37816.019

The following source data is available for figure 3:

**Source data 1.** Values of concentrations of six short chain fatty acids.
DOI: https://doi.org/10.7554/eLife.37816.020

opportunity in future studies to better understand the basis of the T1D sex dimorphism in NOD mice before puberty.

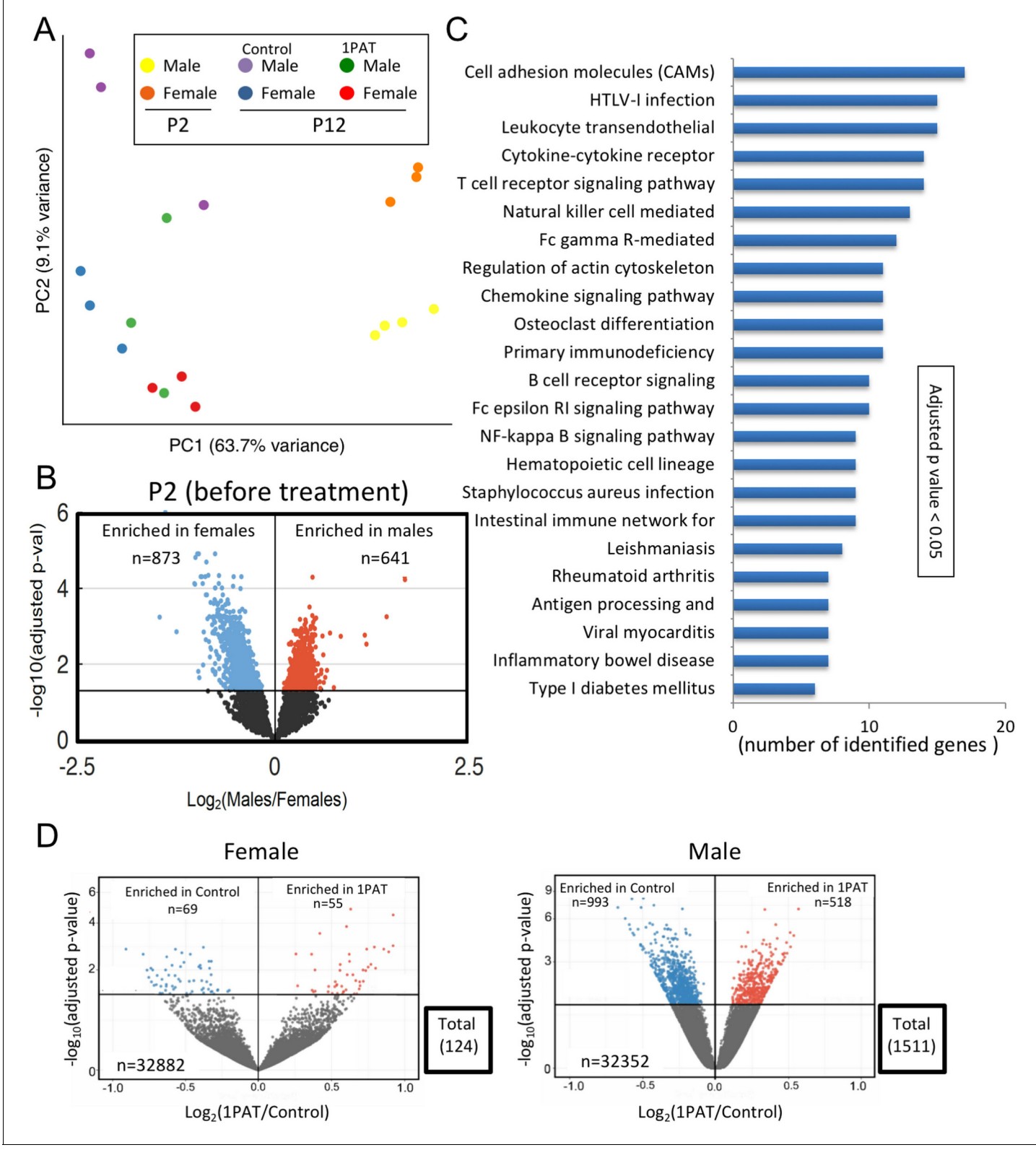

**Figure 4.** 1PAT effects on ileal gene expression from P2 to P23, based on RNA-Seq analysis. (A) Bray-Curtis analysis of ileal gene expression of pups at P2 and P12 (2 days after PAT), represented by PCA. All differences between expression at P2 and P12 (in both 1PAT and control mice) were significant in both males and females (p<0.05). (B) Differential ileal gene expression between P2 males and females; volcano plot indicates 1514 genes with significantly different expression. DESeq2 difference visualization maximizes at 15 (-log$_{10}$ (adjusted p values)). (C) Ileal KEGG pathways altered by 1PAT

*Figure 4 continued on next page*

*Figure 4 continued*

in P12 males; 23 significant pathways (adjusted p<0.05, after Benjamini-Hochberg correction), bars indicate numbers of differentially affected genes/pathway. (D) Differential ileal gene expression in 1PAT and control P23 females (left) and males (right) shown by volcano plot. In females, 124 genes (55 up-, 69 down-), and in males, 1511 genes (518 up-, 993 down-) were significantly differentially expressed, a 12.2-fold difference between the sexes. n = 4, 3, and 6–7 mice per group, at P2, P12, and P23, respectively. [See also *Figure 4—figure supplements 1–3*, and *Supplementary files 3* and *4*].
DOI: https://doi.org/10.7554/eLife.37816.021

The following figure supplements are available for figure 4:

**Figure supplement 1.** Differential ileal gene expression maturation between P2 and P12 in control and PAT mice, as shown in volcano plots.
DOI: https://doi.org/10.7554/eLife.37816.022

**Figure supplement 2.** Unsupervised hierarchical clustering of significantly differential genes in P23 1PAT and control mice as assessed by RNA-Seq, and shown on heat maps.
DOI: https://doi.org/10.7554/eLife.37816.023

**Figure supplement 3.** Effect of 1PAT on ileal gene expression profiles in 1PAT and control male mice at P42, showing an unsupervised hierarchical clustering of all significantly differential genes.
DOI: https://doi.org/10.7554/eLife.37816.024

## Microbial perturbations affect the maturation of early-life gene expression in the ileum

Next, we examined mice at P12 to assess the maturation of gene expression (defined as the significant differences compared with P2), and the effects of PAT on that maturation. At a global level, the expression profiles were significantly different between PAT and control in both males and females (*Figure 4A*); thus by P12, the antibiotic effects on the microbiome were already being transduced into the tissues. As expected with normal development, expression of several thousand genes significantly changed in the control mice from P2 to P12, however, in the mice receiving PAT between P5–10, ~17% of the differences were lost in males and ~21% in females (*Figure 4—figure supplement 1*). KEGG analysis of the male mice highlighted significant changes between PAT and control in cell adhesion molecules, cytokine–cytokine receptor interaction, T cell receptor signaling, B cell receptor signaling, intestinal immune network for IgA production, and leukocyte transendothelial migration. (*Figure 4C*).

By P23, we found striking differences between males and females; in males, 1511 ileal genes showed significant differential expression between PAT and control vs. only 124 in females, a 12.2-fold difference (p<0.001) (*Figure 4D*, *Figure 4—figure supplement 2* and *Supplementary file 3*). Thus, in a functional sense, the female ileum had greater resilience against the same microbiological perturbation (*Figure 2—figure supplement 9*). In the males, the genes that had significantly reduced expression included *Nos2*, *Saa1*, *Runx1*, and *Muc4*, all involving host defenses (see below). By P42, nearly half of the significant differences in males between 1PAT and controls were lost; nevertheless, 32 days after the antibiotic exposure had ended, there remained abnormal expression of hundreds of ileal genes (*Figure 4—figure supplements 2* and *3*). Thus, RNA-Seq studies indicated a broad effect of the microbiome changes on the maturation of early-life intestinal gene expression, with asymmetric effects in males and females, consistent with their differential development of accelerated T1D later in life.

## Maturation of ileal genes and pathways related to immunity and inflammation

Since RNA-Seq provided a global assessment of ileal gene expression, we next sought to focus on a specific subset of 547 genes related to inflammation and immunity as captured on the NanoString nCounter mouse immunological assay (*Figure 5*). First, Bray-Curtis distance matrix analysis indicated strong and directional age-related effects from P12 to P42 (*Figure 5A*), independently confirming and extending the RNA-Seq findings indicating maturation of gene expression. Using the time-series of samples, we compared the age-associated normal (control) immune/inflammatory gene expression with that after the brief PAT exposure. The period from P12 to P23 was more developmentally dynamic than that from P23 to P42, consistent with the broader RNA-Seq findings (*Figure 5—figure supplement 1* top panel). However, although the number of genes with changed expression status in the 1PAT mice was similar to control (*Figure 5—figure supplement 1* bottom panel), many of the maturing genes differed. As such, we defined five classes of ileal immune expression maturation

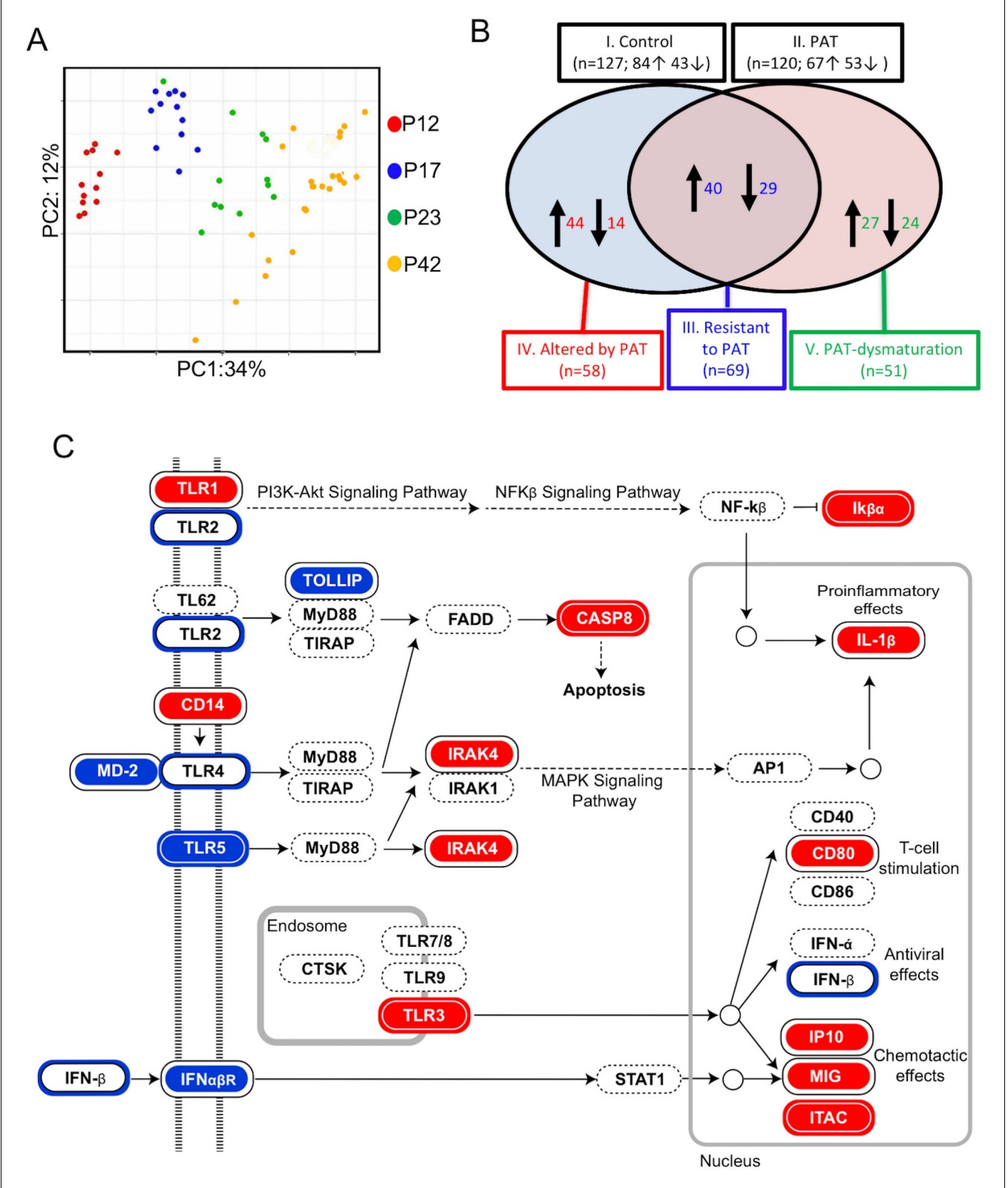

**Figure 5.** 1PAT effects on ileal immune gene expression in NOD mice, as evaluated by NanoString. (**A**) Bray-Curtis analysis of ileal gene expression of pups, over time (P12, P17, P23 and P42). (**B**) Comparison of ileal gene expression in1PAT and Control NOD males, defining five classes of genes maturing between P12 and P42. I. Maturing in Control mice; II. Maturing in 1PAT mice; III. Maturation that is resistant to 1PAT; IV. Maturation that is altered by 1PAT; and V. Maturing exclusively in1PAT mice (dysmaturation). Arrows indicate direction of maturation of expression from P12 to P42, and

*Figure 5 continued on next page*

*Figure 5 continued*

number of genes affected. (C) Differences in Toll-like receptor signaling pathway genes between 1PAT and control males maturing from P12 to P23, based on KEGG pathway analysis. Boxes filled with blue indicate that gene expression significantly decreased in controls during maturation from P12 to P23; filled with red indicate significant increases; Blue and red circles indicate significant decreases or increases, respectively, in the 1PAT mice. Black circles indicate that the significant change in controls was lost in 1PAT mice. All significance testing performed by Fisher's exact test with the Benjamini–Hochberg correction. [See also *Figure 5—figure supplements 1–3*].
DOI: https://doi.org/10.7554/eLife.37816.025

The following figure supplements are available for figure 5:

**Figure supplement 1.** PAT effect on the maturation of ileal immune gene profiles in male NOD mice, as assessed by NanoString nCounter assays and shown on heat maps.
DOI: https://doi.org/10.7554/eLife.37816.026

**Figure supplement 2.** Differences in NOD-like receptor signaling pathway between PAT and control males maturing from P12-P23.
DOI: https://doi.org/10.7554/eLife.37816.027

**Figure supplement 3.** Differences in Th17 cell differentiation pathway between PAT and control males maturing from P12 to P42.
DOI: https://doi.org/10.7554/eLife.37816.028

(*Figure 5B*), including genes that significantly matured in: (I). Controls; (II). after 1PAT; (III). maturation was resistant to 1PAT; (IV). maturation was altered by 1PAT; or (V). represented a new pattern of 1PAT-induced maturation (dysmaturation). Such a classification strategy may have general utility in interpreting data from global genetic approaches in other models involving developmental perturbations.

Using these individual gene-level expression differences to understand which KEGG pathways were differentially affected by 1PAT permitted identification of important innate pathways, for example, Toll-Like Receptor (TLR) signaling (*Figure 5C*). Although the changes induced by PAT were widespread, analyses also indicated effects, for example, on pathways involved in NOD-like receptor signaling and Th17 cell differentiation (*Figure 5—figure supplements 2* and *3*).

## Microbial perturbations affect specific genes related to host responses

Next, to confirm and extend these broad RNA-Seq and NanoString observations, we explored individual genes of particular interest at the host-microbial interface. Using qPCR studies to validate the global findings, we found that in the PAT mice, expression was reduced for genes encoding four of the five transcription factors regulating the inducible nitric oxide synthase (*Nos2*), for *Nos2* itself, and the related *Calm3* (*Figure 6A*). *Runx1*, an early life transcription factor showed reduced expression with PAT, as did two of its downstream genes (*Foxp3* and *Cd3g*) that are involved in the development of adaptive immunity (*Figure 6B*), but effects on the innate *Saa* genes were bimodal over time (*Figure 6—figure supplement 1A*). There was marked reduction of expression of the two major genes involved in intestinal mucin synthesis (*Muc2* and *Muc4*) (*Figure 6—figure supplement 1B*). *Ido1*, encoding indoleamine 2,3-dioxygenase, the key enzyme catalyzing tryptophan catabolism along the kynurenine pathway and having a major role in immune modulation by mediating T-cell inhibition depending on bone marrow stromal cell activation (*Meisel et al., 2004*; *Munn and Mellor, 2013*), showed reduced expression with PAT (*Figure 6—figure supplement 1C*). The RT-qPCR studies confirmed the gradual normalization of most but not all of the candidate genes (*Figure 6A,B* and *Figure 6—figure supplement 1*). In total, these studies provide evidence that the 1PAT-altered microbiota interferes with immune pathways during a critical developmental window.

## PAT regulation of histone post-translational modification states

Based on the strong 1PAT-induced differences in early life intestinal gene expression, including transcription factors, we next considered an epigenetic basis for the changes. We examined global histone post-translational modification (PTM) states to assess whether differential chromatin regulation contributes to the enhanced T1D phenotypes in the 1PAT-exposed male mice. We focused on ileum and liver, based on the 3PAT-induced sex-specific differences in ileal gene expression (*Livanos et al., 2016*), on the effect of the early-life antibiotic exposure on gene expression and metabolism we report here, and on prior identification of robust gut microbiota-driven global changes in intestinal and hepatic histone PTM states (*Krautkramer et al., 2016*). Measuring 65

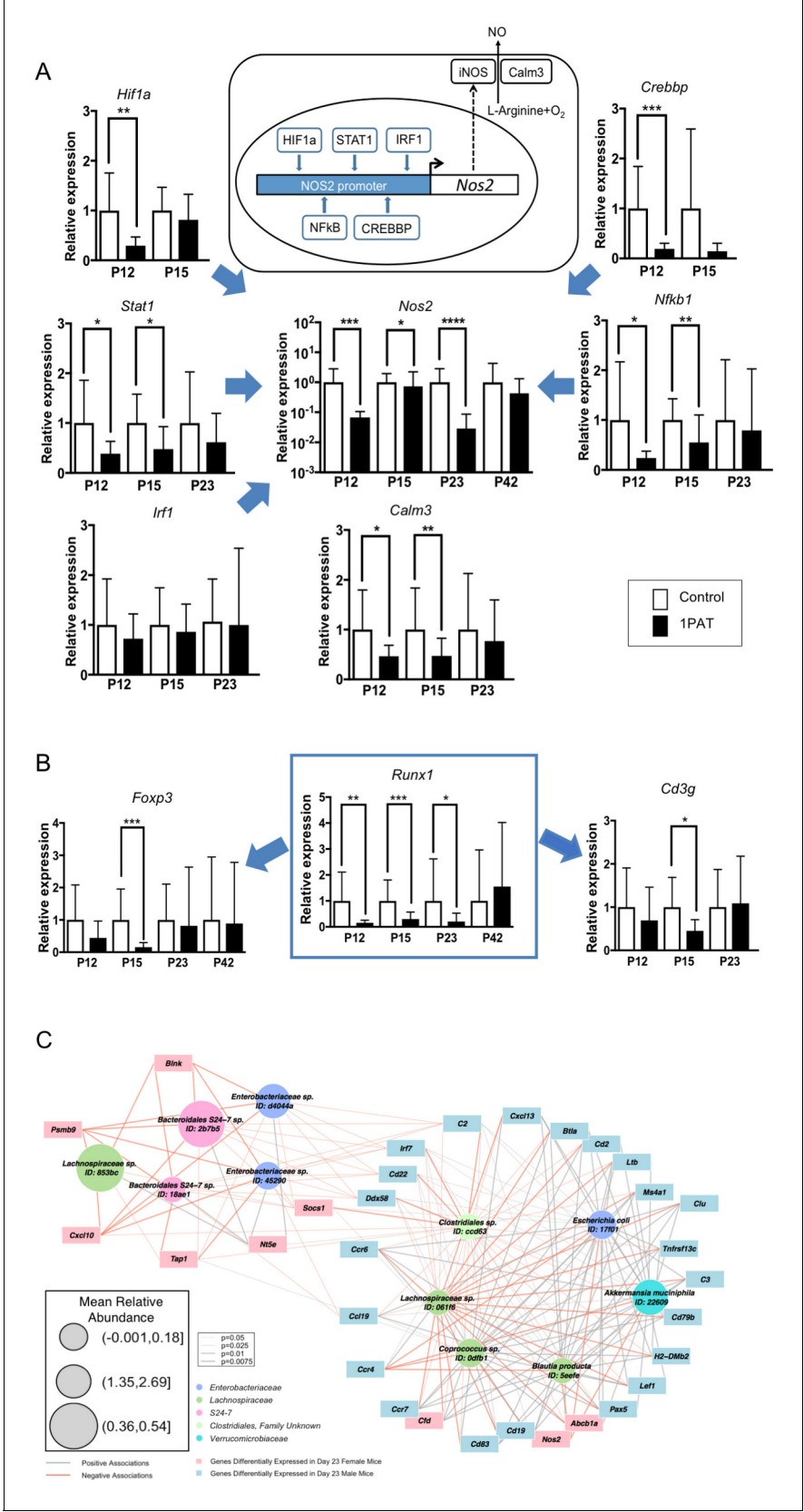

**Figure 6.** Differential effects of 1PAT on transcription of specific ileal genes in P12 to P42 mice, as determined by RT-qPCR, and microbe-host gene expression interactions. (**A**) Transcription of the inducible nitric oxide synthase gene *Nos2* and its upstream transcription factor genes *Stat1*, *Nfkb1*,

*Figure 6 continued on next page*

*Figure 6 continued*

*Hif1a*, *Crebbp*, and *Irf1*, as well as *Nos2* partner *Calm3*. (B) Transcription factor *Runx1* and its downstream immune genes *Foxp3* and *Cd3g*. n = 10 ~22 samples per group, and mean values with STDEV were shown. Statistical significance determined by the Mann Whitney test. *p<0.05; **p<0.01; ***p<0.001; and ****p<0.0001. (C) CompPLS framework selecting significant associations identified differential taxa and ileal gene expression in P23 male mice. The 43 immune genes profiled by NanoString were selected as the targeted significantly differential genes, and the model found 31 (72%). Red and gray edges indicate negative and positive model-selected associations, respectively. Edge width in the network indicates statistical significance as derived from empirical p values after bootstrapping. Node sizes are scaled to the mean relative abundance for each taxon, and color-coded by OTU family. [See also *Figure 6—figure supplements 1–3* and *Supplementary files 3* and *4*].

DOI: https://doi.org/10.7554/eLife.37816.029

The following source data and figure supplements are available for figure 6:

**Source data 1.** Relative expression values for 10 ileal genes.
DOI: https://doi.org/10.7554/eLife.37816.033
**Figure supplement 1.** Differential effects of PAT on *Saa*, mucin genes, and *Ido1* expression, as determined by RT-qPCR.
DOI: https://doi.org/10.7554/eLife.37816.030
**Figure supplement 2.** Significant associations of intestinal taxa with 31 strongly differentially expressed ileal genes in P23 PAT mice.
DOI: https://doi.org/10.7554/eLife.37816.031
**Figure supplement 3.** Changes in ileal *Egf* expression with age, either alone or as a ratio to *Egfr*.
DOI: https://doi.org/10.7554/eLife.37816.032

acetylated and methylated histone PTM states at P23, we found differential effects between PAT and control (*Supplementary file 5*), but notably greater effects in males than females (*Figure 7*), consistent with both the differences in gene expression and the T1D phenotypic enhancement. The PAT effects on hepatic chromatin were more robust than for the ileum (*Figure 7A*), with greater responses again in males than in females (*Figure 7B*). Further, the RNA-Seq studies from the P23 male mice showed dysregulated expression of histone modifying genes including anti-silencing histone chaperone *Asf1b*, and HADC-binding transcription factors *Mier1* and *Mier3*. These results, in conjunction with the altered SCFA production (affecting transcription of multiple histone-modifying genes (*Krautkramer et al., 2016*; *Terova et al., 2016*), provide a further connection between the PAT-altered microbiota and the altered intestinal gene expression.

## PAT-induced changes in adaptive immunity

Given that T1D involves immune-mediated destruction of pancreatic islets, we next asked whether the 1PAT-altered microbiome and its downstream effects on metabolism and innate immunity had differential effects on adaptive immune mediators. First, we examined whether there might be differential B-cell effects between PAT and control Assessing fecal IgA levels, we found consistent decreases at least to P70 in both the PAT-exposed males and females (*Figure 8A*), with findings paralleling those observed in C57BL/6 mice (*Ruiz et al., 2017*). To assess changes in other immunological loci, lymphocytes of the pancreas, and spleen were immunophenotyped in P42 PAT and control mice, after many innate differences had normalized, but still prior to the insulitis observed at P70. In the spleen, there were increased frequencies of $CD62L^+$ $CD4^+$ and $CD62L^+$ $CD8^+$ T cells (*Figure 8B*), indicating that the PAT-induced changes systematically increase the pool of naïve T cells in early life and decrease T cell differentiation. The 1PAT mice had greater numbers of B cells in the spleen and pancreas compared with controls (*Figure 8B*). These results indicate that following PAT, there are altered frequencies of adaptive immune cells downstream of the intestine in the spleen and in the target organ (pancreas), preceding T1D development. Differences in absolute count of lymphocytes were not observed in T cells or B cells, indicating there were no changes in lymphocyte quantity (data not shown). However, the change in frequency of naive and activated T cells suggests that the observed differences in the systemic differentiation of these cells does not impact total cellularity.

## PAT exposure creates an alternative network of microbial-host gene expression interactions

To more directly examine the relationship of microbiota changes with differential gene expression, we studied 43 ileal immune genes significantly differing in expression at P23 between 1PAT and control (*Supplementary file 6*) in relation to the 15 differentially represented taxa (*Figure 2C*).

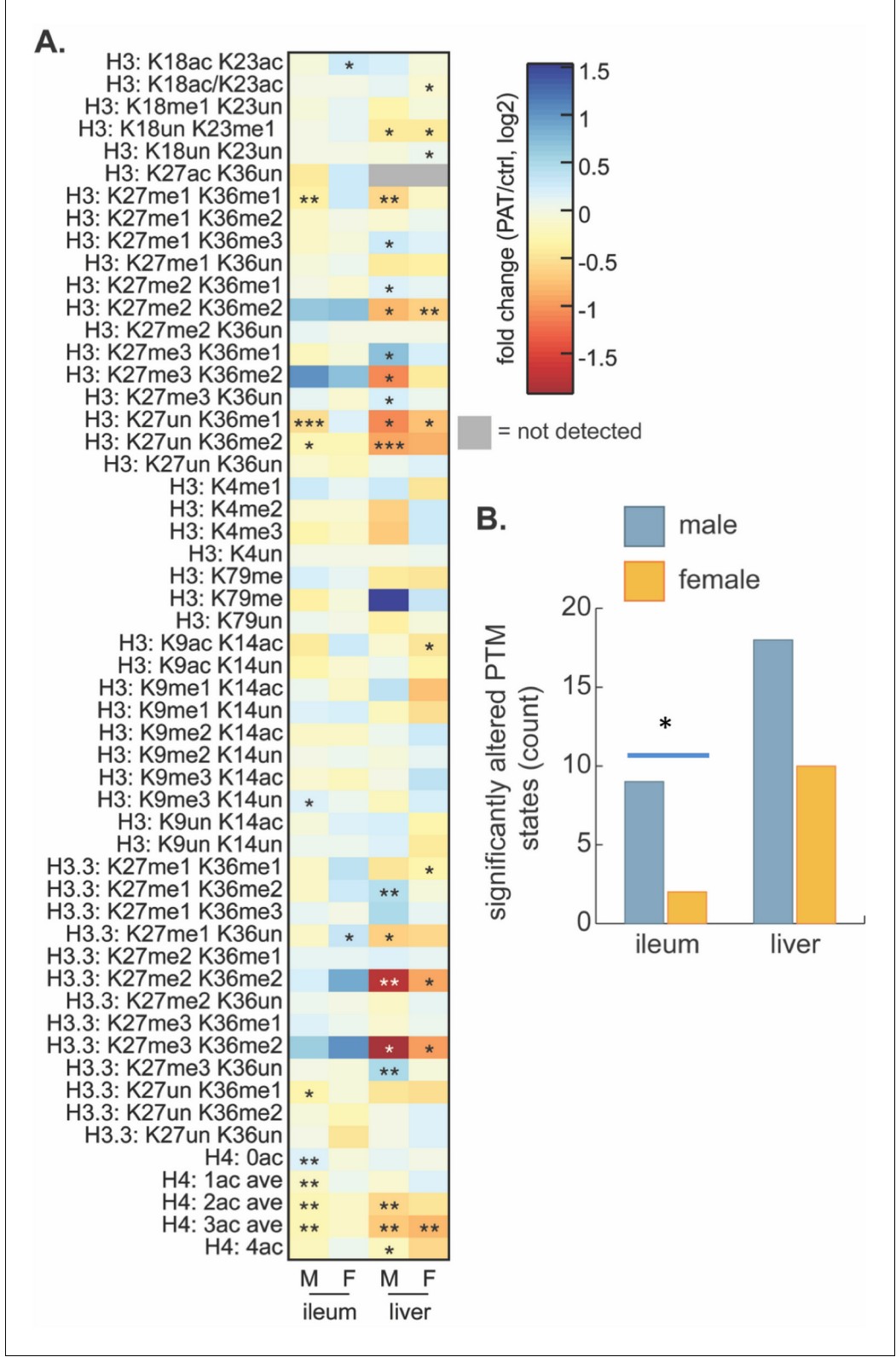

**Figure 7.** Effect of 1PAT on tissue-specific histone PTM state in P23 mice. (**A**) Fold-change ($\log_2$) of 65 unique histone PTM states in male (M) and female (F) ileal and hepatic tissues. Statistical significance determined by Welch's t test (n = 4–5 mice per group). *$p < 0.05$, **$p < 0.01$, and ***$p < 0.001$. (**B**) Summation of significantly altered

*Figure 7 continued on next page*

*Figure 7 continued*

histone peptides in male and female ileum and liver; significance determined by Chi square analysis. *p<0.05. [See also **Supplementary file 5**].

DOI: https://doi.org/10.7554/eLife.37816.034

CompPLS modeling (**Ramanan et al., 2016**) indicated that 31 (72%) genes differentially expressed after 1PAT exposure were significantly correlated with the differential taxa (**Figure 6C**), creating a model of the interactions between the dominant differentiating microbial taxa and the affected intestinal genes. The patterns of connections between taxa and ileal gene expression in males and females markedly differed (**Figure 6—figure supplement 2**), consistent with the phenotypic differences. Particular taxa had relationships across a conserved group of specific families of host genes (**Figure 6—figure supplement 2**). This analysis linked the effects of 1PAT on the microbiota with the downstream ileal gene expression, and separating the males and females.

## Discussion

Our prior studies in C57/Bl6 mice that show that PAT has no effect on gene expression in the ileum in the absence of a microbiota (**Ruiz et al., 2017**) indicate that the ileal gene expression effects we observed in the PAT-exposed mice were due to the microbiota/metagenemic shifts and were not direct antibiotic effects. We now show in NOD mice that the gut microbiome was substantially remodeled by a single early-life PAT exposure, losing diversity without recovery over the entire window relevant to the development of auto-immunity, and with selection for taxa that may be highly metabolically active. One hypothesis to explain the heightened auto-immunity is that reduced diversity of the gut microbiome compromises the ability to control metabolically active opportunistic bacteria. Such organisms, as represented by the short list we identified (**Figures 2C** and **7**), might dysregulate early-life immune responses. Alternatively, the loss of particular beneficial organisms that participate in normal host metabolic and immune maturation, for example, through stimulating mucus production, might trigger the pathogenic pathway. These hypotheses are not exclusive, and rather may be ecologically linked, as suggested by our network model (**Figure 9**).

This study revealed a small consortium of pathobionts (*Enterococcus, Blautia*, and Enterobacteriaceae species) with consistently increased relative abundances in the PAT-exposed male mice that developed accelerated T1D. *Akkermansia*, a taxon that has been inversely correlated with obesity, inflammation, and metabolic syndrome (**Cani and de Vos, 2017**), also is present in increased relative abundance in PAT-exposed male mice. The significant overrepresentation of these four taxa in both 1PAT and 3PAT mice confirms their prior disease-association using the same model in a different mouse facility (**Livanos et al., 2016**), and they also have been linked to pathological processes in other intestinal microbiome studies (**Hänninen et al., 2018**; **Kim et al., 2017**; **Minter et al., 2017**): (i), *Enterococcus* species, common in both the human and murine gut, secrete products that alter host immune responses through NFkB/TLR pathways (**Tien et al., 2017**), perturb other commensals (**Fisher and Phillips, 2009**), translocate to the liver, inducing autoimmunity, including autoantibodies (**Manfredo Vieira et al., 2018**), and also induce infiltration of CD42$^+$ MPO$^+$ cells into the rat pancreas (**Korsgren et al., 2012**); (ii), Children who later developed T1D have had altered gut microbiota representation of *Blautia* (**Murri et al., 2013**; **Qi et al., 2016**); (iii), *Akkermansia mucinophila*, intestinal mucin-degrading bacteria, may modulate host innate immunity (**Derrien et al., 2011**, **2004**), including endocannabinoids, affecting inflammation, gut barrier permeability and peptide secretion (**Everard et al., 2013**). Long-term exposure of older female NOD mice to vancomycin identified *Akkermansia* as a T1D-protective taxon (**Hansen et al., 2012**), but that experiment differed substantially from ours; (iv), in our model, an Enterobacteriaceae taxon other than *E. coli* was induced by 1PAT across P21 to P49, consistent with identification of an unclassified Enterobacteriaceae member as the most significantly increased taxon in T1D children compared to healthy controls (**Soyucen et al., 2014**). Enterobacteriaceae induce innate immune responses via Toll-like receptor 4 (TLR4) (**Tapping et al., 2000**; **Vasselon and Detmers, 2002**), which was dysregulated by PAT perturbation (**Figure 5C**). Enhanced TLR responses to the Enterobacteriaceae could propagate immunopathology, consistent with microbiota regulating T1D development through MyD88-dependent TLRs

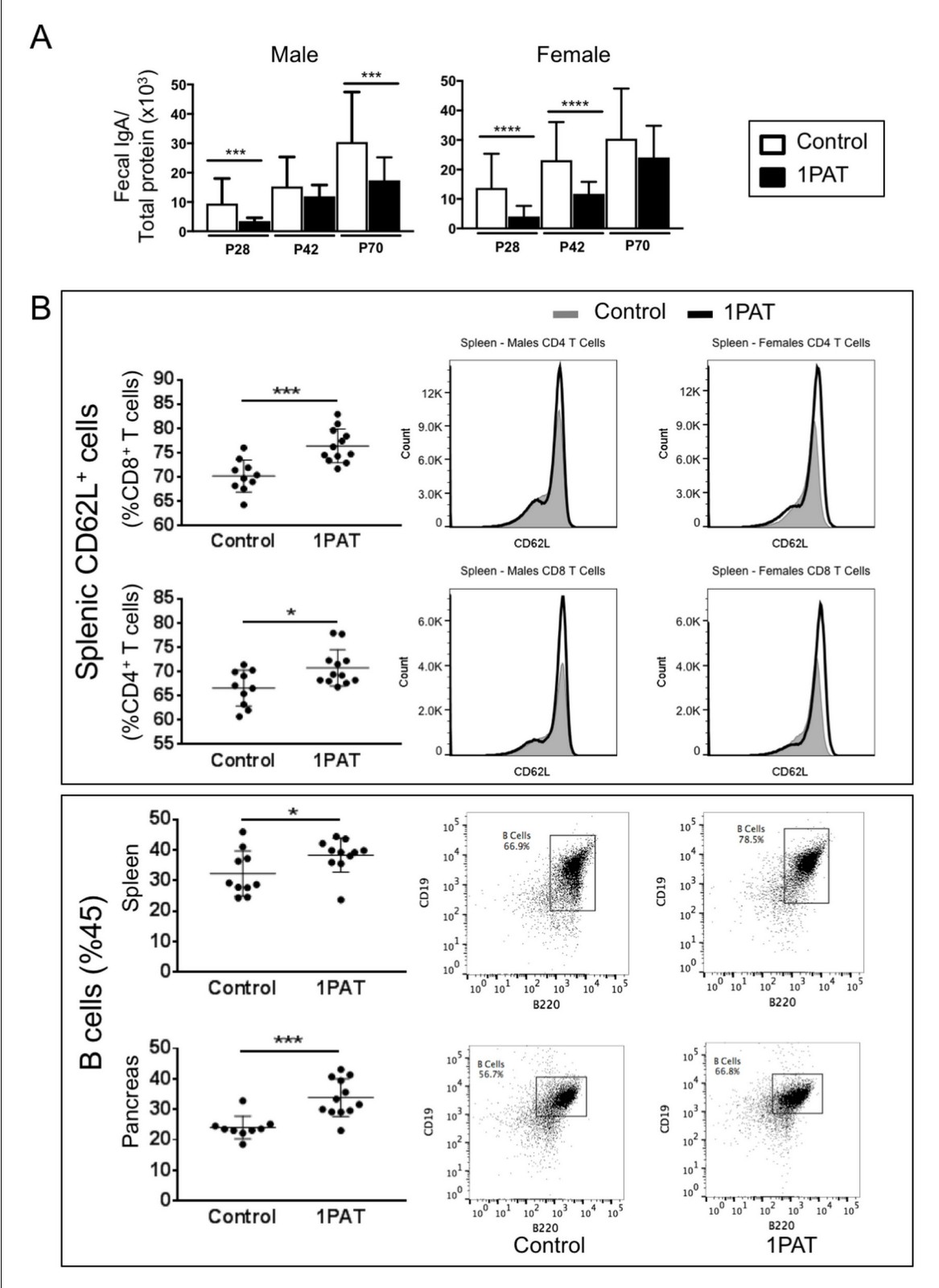

**Figure 8.** Effect of 1PAT on host immune responses. (**A**) Fecal IgA levels in male and female NOD mice in 1PAT or control groups from P28 to P70, as determined by ELISA. Statistical significance, comparing 1PAT and control (n = 20–25 samples per group) was determined by the Mann Whitney test. *p<0.05; **p<0.01; ***p<0.001. (**B**) Flow cytometric analysis of immune cells of P42 NOD mice. 1PAT treatment significantly increased proportions of splenic memory T cells (both CD4⁺ CD62L⁺ and CD8⁺ CD62L⁺ cells) and B cells, especially in females (data not shown), and increased the proportion of

*Figure 8 continued on next page*

*Figure 8 continued*

B cells in the pancreas (in both males and females). Statistical significance, comparing 1PAT and control (n = 10–12 samples per group) was determined by the unpaired T test. *p<0.05; **p<0.01; ***p<0.001.

DOI: https://doi.org/10.7554/eLife.37816.035

The following source data is available for figure 8:

**Source data 1.** IgA ELISA and flow cytometry data.

DOI: https://doi.org/10.7554/eLife.37816.036

(*Burrows et al., 2015*). Biosynthetic gene cluster (BGC) analysis from the metagenomic sequencing revealed several enriched secondary metabolite pathways in 1PAT mice which map to an *Enterococcus* polysaccharide pathway, which could affect host immune responses (*Teng et al., 2002*; *Xu et al., 1998*), and also the Enterobacteriaceae siderophores, aryl polyenes, and NRPS family products (*Figure 2E*). Bacterial siderophore and related transition-metal scavenging responses are often associated with pathobiont organisms and activity, and have been shown to be important mediators of bacterial community structure and of bacterial persistence within human hosts during infection. Furthermore, certain bacterial siderophores interact with the immune defense and

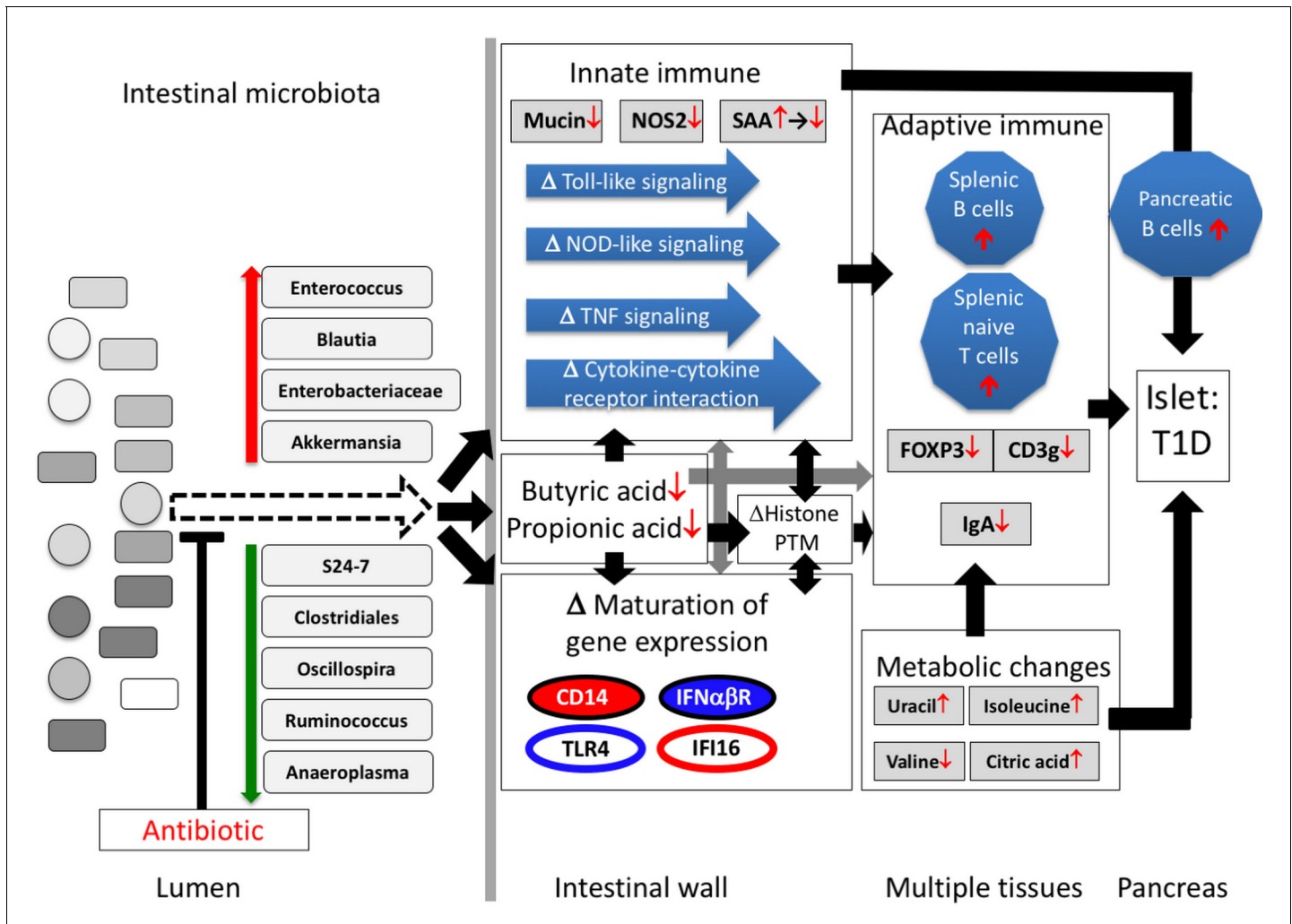

**Figure 9.** Proposed model of how a single early-life antibiotic course can perturb the microbiota/metagenome/metabolome leading to dysregulated intestinal innate and adaptive immunity to accelerate T1D in NOD mice. For differences in maturation of gene expression, see *Figure 5C* legend.

DOI: https://doi.org/10.7554/eLife.37816.037

immunomodulatory protein Lipocalin 2 in mice and humans, and could be relevant for normal immune development and tolerance (*Berard et al., 2012*; *Raffatellu et al., 2009*). *Enterobacteriaceae* (e.g. *E. coli*) aryl polyenes protect bacteria from oxidative stress from immune cells during colonization (*Cimermancic et al., 2014*), which could interfere with bacteria-host interaction and immune development. NRPS family products are broad in structure but include metabolites with demonstrated anti-inflammatory and immunosuppressant activity such as cyclosporin A (*Felnagle et al., 2008*). The small pathobiont consortium found to bloom in this study has many requisite features for T1D immunopathogenesis.

We also identified S24-7, *Clostridiales*, *Oscillospira*, *Ruminococcus,* and *Anaeroplasma* as taxa that were potentially protective against enhanced T1D. Unclassified Bacteroidales of the S24-7 family were associated with T1D protection in this study, in our prior 3PAT model in male NOD/ShiLtJ mice (*Livanos et al., 2016*), and in studies in female NOD/BomTac mice (*Krych et al., 2015*). The underrepresented *Anaeroplasma* in PAT is consistent with the recent report that in NOD mice, this genus also was under-represented in the gut of those with low-diabetes-frequency compared with those at high-diabetes-frequency (*De Riva et al., 2017*). Prior findings related to *Clostridia*, *Oscillospira*, and *Ruminococcus* were less consistent (*Krych et al., 2015*). In total, our findings support the hypothesis that by selectively diminishing particular (beneficial) taxa, PAT exposure permits emergence of a less diverse microbiota (*Figure 2B*), dominated most likely by highly metabolically active host-interactive taxa.

Male and female NOD mice, even from the same litter, have substantially different T1D rates (*Markle et al., 2013*), and differing immune response profiles (*Bao et al., 2002*). Sex-specific differences in intestinal gene expression profiles present at P2, prior to any antibiotic exposure, provide one explanation for the differential disease rates. We found that the signals from the 1PAT-altered microbiome are differentially transduced in the intestinal tissues of sibling male and female mice, despite their same mothers, diet, mouse facility, and microbiome composition. The very high rate of T1D development in the control females may have fully saturated the major pathogenetic pathway. The histone, RNA-Seq, and NanoString-based analyses clearly indicate that young male and female NOD mice differ in their responses to the same PAT-induced gut microbiome perturbations. Our studies of early life sex-specific ileal gene expression differences provide opportunities to better understand the sex-based differences in autoimmune disease pathogenesis, extending other recent work (*Thion et al., 2018*).

Exposure to 1PAT decreased the total numbers of genes maturing in both early life males and females, disproportionately affecting those involved in innate and adaptive immune pathways, including those in Toll-like receptor (TLR) signaling, NOD-like receptor signaling, and Th17 cell differentiation; each of these pathways have been implicated in prior T1D studies (*Burrows et al., 2015*; *Lien and Zipris, 2009*; *Livanos et al., 2016*; *Wen et al., 2008*). Th17 roles in NOD mice are not clear-cut; blocking Th17 generation successfully inhibited T1D development (*Lee et al., 2013*), while inducing Th17 cells with particular gut microbiota protected from T1D development in the same animal model (*Bedoya et al., 2013*). Our study revealed that multiple components of the Th17 pathway are dysregulated in opposing directions, indicating the complexity of this immunological pathway in the disease phenotype.

The pro-inflammatory serum amyloid protein A (SAA) is significantly induced in T1D patients (*Zhi et al., 2011*). The biphasic response of *Saa1* to the microbiota changes that we observed may reflect the dominance of highly host-interactive organisms early, and the altered innate signaling pathways later. Perturbation of the mucin layer, a major barrier to the interaction of the microbiota with intestinal cells has been associated with intestinal diseases (*Crost et al., 2016*; *Rokhsefat et al., 2016*; *Tailford et al., 2015*). *Nos2* is critical for cellular signaling and immune defense responses (*Bogdan, 2015*), since NO activates Th1 phenotypic responses, regulates anti-inflammatory pathways, and mediates tissue restoration (*Wink et al., 2011*). *Nos2* expression, down-regulated early in 1PAT recipients, may be a good sensor of intra-luminal events, transducing the microbial signals to downstream host functions (*Atarashi et al., 2015*).

Given that binding of Epidermal Growth Factor (EGF) to the EGF receptor (EGFR) subsequently activates multiple signaling pathways—including PAR2, TLR, Ras/MARK, PI3K/AKT and STAT—which regulate the intestinal barrier and permeability (*Good et al., 2012*; *Tang et al., 2016*; *Wang et al., 2015*), we examined their maturation with age, and the effects of sex and PAT exposure on expression. *Egfr* expression was tightly conserved across the 47 specimens examined from

P2 to P23 (relative Mean expression $4.37 \pm 0.62 \times 10^{-5}$). Very young (P2) males had significantly higher levels of *Egfr* expression than young females. Subsequently, all rates normalized by P12 and thereafter (*Figure 6—figure supplement 3A*). For *Egf*, levels did not significantly differ according to treatment or sex (Mean $1.95 \pm 0.58 \times 10^{-6}$), but diminished between P12 and P23 (*Figure 6—figure supplement 3B*). Findings were similar if *Egf* was examined as a ratio in relation to *EgfR* (*Figure 6—figure supplement 3C*). Thus, we observed that there is a very early in life (P2) increased level of *Egf* in males, with rapid normalization, but that *Egf* levels significantly fell with weaning. *Egfr* is an example of a gene with sex-specific early life expression differences. In contrast, *Egf* is a gene whose expression in the ileum undergoes age-related changes, but which are independent of both sex and treatment effects [Class III (PAT-resistant) in *Figure 5B*]. Given that *Egfr* affects tight junctions and adherence in the colon (*Basuroy et al., 2005*), the fall at weaning could explain later bacterial translocation and downstream immunological effects. In total, the altered expression of the identified/validated 1PAT-dysregulated genes may participate in the subsequent intestinal immune system dysmaturation.

The histone modification data are consistent with the dysregulation of ileal gene expression in this and our prior study (*Livanos et al., 2016*). The histone data also are in line with sexual dimorphism in how the gut microbiota communicate early-life environmental exposures to host chromatin (*Figure 7B*), consistent with both the disease risk dimorphism (*Yurkovetskiy et al., 2013*), and the early-life transcriptional phenotypes we observed. To link 1PAT-induced changes in global histone PTMs to genomic loci and ultimately to downstream effects on gene expression, both the GEO (Gene Expression Omnibus) (*Barrett et al., 2013*; *Edgar et al., 2002*) and ENCODE (ENCyclopedia Of DNA Elements) (*ENCODE Project Consortium, 2012*) repositories were mined for publicly available ChIP-seq data sets targeting either histone H3 K36me2 or acetylated histone H4, and most specifically histone H4 K16ac. These modifications were selected for further analysis due to the relatively high magnitude of differences and sex-specificity of responses at these sites (*Figure 7*, *Supplementary file 5*). For histone H3, the variant histone H3.3, and histone H4 provide an unsupervised and global assessment of common PTMs. In the male 1PAT ileum, there was a modest but statistically significant decrease in acetylation of histone H4 and a concomitant increase in abundance of unmodified histone H4 compared to controls, suggesting that 1PAT leads to a net loss of acetylated H4. These results are consistent with our previous observations in the C57BL/6 proximal colon that the gut microbiota induces histone acetylation (*Krautkramer et al., 2016*), suggesting that 1PAT exposure decreases key microbial populations important for this chromatin response. There also were significant decreases in H3 K27me1 K36me1, H3 K27un K36me1, and H3 K27un K36me2 peptides in 1PAT ileum relative to controls (*Figure 7A*). Importantly, and in contrast, the effects of 1PAT in the female ileum were minimal (*Figure 7B*). Coupled with the differential gene expression asymmetry between male and female mice, both *a priori* (P2) and PAT-induced, these studies link epigenetic changes with the differential disease phenotypes. Similar to the chromatin signatures in male ileum, the 1PAT exposure resulted in hepatic hypoacetylation of histone H4 and hypomethylation of histone H3 K36-containing peptides in males and females and a concomitant increase in the abundance of peptides containing highly methylated H3K27 (*Figure 7A*). Acetylation of histone H4 and methylation of H3K36 are both associated with active transcription, whereas methylation of H3K27 is associated with transcriptional repression (*Pasini et al., 2010*; *Robinson et al., 2008*; *Zhao and Garcia, 2015*). Other search parameters explored include limiting data to those samples from either mouse or human origin and either intestine or liver tissues or cell lines (including primary cultures or transformed cell lines). However, this search returned no publicly available data sets that met these criteria, precluding further analyses.

Since SAA can induce expression of *Jmjd3* (*Kdm6b*), a histone H3 lysine 27 (H3K27) demethylase, reducing H3K27 trimethylation (*Yan et al., 2014*), *Saa* dysregulation by PAT-induced microbiome perturbation may alter epigenetic status in early life (*Figure 7* and *Figure 6—figure supplement 1*). H3K27 methylation is associated with repression of both *Runx1* through enhancer of zeste homolog 2 EZH2 (*Takayama et al., 2015*), and iNOS (*Dreger et al., 2016*). Thus, our observation that histone H3K27me3K36me2 (K27 trimethylation) was > 2 fold up-regulated by 1PAT while histone H3K27un (unmodified) was significantly down-regulated (*Figure 7A*) is consistent with the RNA-Seq and RT-qPCR analyses showing early up-regulation of *Saa* and down-regulation of *Nos2* and *Runx1* in the 1PAT male mice (*Figure 6A,B*, *Figure 6—figure supplement 1A*, and *Supplement file 3*).

Finding 1PAT-reduction of cecal butyric acid confirmed our prior result with 3PAT (*Livanos et al., 2016*). By inhibiting histone-modifying genes in intestinal cells, SCFAs including butyrate, affect epigenetic status and thus gene expression (*Fellows et al., 2018*; *Schilderink et al., 2013*); the transcriptional analyses we performed provide direct evidence of such dysregulation. Diminished butyrate levels are consistent with the decreased ileal *Runx1* and *Foxp3* expression that we observed, which could differentially shift T-helper cell maturation away from Treg-cells (*Furusawa et al., 2013*; *Geuking et al., 2013*; *Smith et al., 2013*). Our analyses show that antibiotic-induced gut microbiome remodeling reduced intestinal SCFA levels, dysregulated histone PTM status, and repressed mucin genes- all of which contribute to perturbing innate intestinal immunity development. Since butyrate and propionate both increase histone H3 and H4 acetylation and H3 methylation, affecting the *Muc2* promoter and increasing *Muc2* mRNA levels (*Burger-van Paassen et al., 2009*), the low levels we observed are consistent with the decreased *Muc2* expression.

In total, our investigation suggests the following model to explain the 1PAT effects on T1D in the male NOD mice (*Figure 9*). Antibiotic administration in early life selected for particular intestinal microbial populations, continuing weeks after the antibiotic stopped, including small groups of significantly over- and under-represented taxa. Changes in the representation of these taxa, especially those predicted to have highly active metabolism and thus host-interaction, can initiate the primary events in the disease enhancement. The altered microbial populations and their products, including SCFAs, differentially interact with ileal epithelial cells, affecting histone modification, and changing gene expression and its normal maturation. The downstream effects on specific innate genes and pathways and the metabolic changes would then influence how adaptive immunity develops. Together, the broad cascade of events altering the expression of specific host genes and pathways appears sufficient to trigger and accelerate T1D in males.

These studies contribute to a growing body of evidence on the effects of early life antibiotic exposures in mouse models of disease (*Cox et al., 2014*; *Nobel et al., 2015*; *Ruiz et al., 2017*; *Schulfer et al., 2018*), and particularly of T1D (*Candon et al., 2015*; *Livanos et al., 2016*; *Pearson et al., 2016*), and are consistent with some (*Pflüger et al., 2010*; *Yallapragada et al., 2015*), but not all (*Hviid and Svanström, 2009*; *Kemppainen et al., 2017*) epidemiologic studies in humans identifying early life antibiotic exposure as a risk factor for T1D development. This simplified animal model, the taxonomic, metagenomic, and metabolic leads we have identified, and the approach and classification system for gene expression maturation advance understanding of the mechanisms by which gut microbiome perturbations contribute to the pathogenesis of T1D and to other immune-associated diseases.

# Materials and methods

## Key resources table

| Reagent type (species) or resource | Designation | Source or reference | Identifiers | Additional information |
|---|---|---|---|---|
| Strain (Mus musculus) background | NOD/ShiLtJ mice | Jackson Laboratory (Bar Harbor ME) | | |
| Chemical compound, drug | Tylosin tartrate | Sigma-Aldrich, Billerica MA | Cat#T6134-25G | |
| Biological sample | Mouse tissues: liver, pancreas, serum; ileum; colon | This paper | | |
| Biological sample | Mouse microbiota samples: fecal cecal contents, ileal contents | This paper | | |
| Antibody | CD45 BV650 | BioLegend | 30-F11 | |
| Antibody | CD3 AF700 | BioLegend | 17A2 | |
| Antibody | CD8a PacificBlue | BioLegend | 53–6.7 | |
| Antibody | CD4 AF488 | BD BioSciences | RM4-5 | |

*Continued on next page*

*Continued*

| Reagent type (species) or resource | Designation | Source or reference | Identifiers | Additional information |
|---|---|---|---|---|
| Antibody | CD19 PE-Cy7 | BioLegend | 6D5 | |
| Antibody | B220 PerCPCy5.5 | BioLegend | RA3-6B2 | |
| Antibody | CD44 PE-Dazzle 594 | BioLegend | IM7 | |
| Antibody | CD62L BV570 | BioLegend | MEL-14 | |
| Sequence-based reagent | PCR primer pairs | See **Supplementary file 7** | | |
| Commercial assay or kit | FreeStyle Lite meter/blood lgucose test strips | Abbott Diabetes Care Inc., Abbott Park IL | Cat#99073070822 | |
| Commercial assay or kit | PowerLyzer PowerSoil DNA Isolation Kit | MoBio, Carlsbad CA | Cat#12855–100 | |
| Commercial assay or kit | PowerSoil-htp 96 Well Soil DNA Isolation Kit | MoBio, Carlsbad CA | Cat#12955–4 | |
| Commercial assay or kit | PureLink RNA Mini Kit | Invitrogen, Carlsbad CA | Cat#12183020 | |
| Commercial assay or kit | nCounter GX mouse immunology kit | NanoString Technologies, Seattle WA | Cat#XT-CSO-MIM1-12 | |
| Commercial assay or kit | Verso cDNA Synthesis kit | Thermo Scientific, Waltham MA | Cat#AB1453A | |
| Commercial assay or kit | SYBR Green PCR Master mix | Roche, Branchburg NJ | Cat#12183020 | |
| Commercial assay or kit | mouse IgA ELISA kit | Bethyl, Montgomery TX | Cat#E90-103 | |
| Commercial assay or kit | BCA Protein Assay Kit | Thermo Scientific, Waltham MA | Cat#23225 | |
| Commercial assay or kit | Counting Beads | ThermoFisher | CountBright Absolute Counting Beads; C36950 | |
| Software, algorithm | compPLS | DOI: 10.1186/s13073-016-0297-9 | | https://github.com/zdk123/compPLS |
| Software, algorithm | SHI7 | PMID: 29719872 | | https://github.com/knights-lab/shi7 |
| Software, algorithm | 'custom Python and C code' | DOI: 10.5281/zenodo.1208675 | | https://github.com/RRShieldsCutler/clusterpluck |
| Software, algorithm | QIIME2 | doi:10.1186/s40168-018-0470-z | | https://github.com/qiime2/q2-feature-classifier |
| Software, algorithm | *HUMAnN2 v0.9.5* | doi: 10.1371/journal.pcbi.1002358 | RRID:SCR_014620 | http://huttenhower.sph.harvard.edu/humann |
| Software, algorithm | STAR v2.5.2b | doi: 10.1093/bioinformatics/bts635 | | |
| Software, algorithm | *Ingenuity Pathway Analysis* | IPA, QIAGEN Redwood City | RRID:SCR_008653 | http://www.ingenuity.com |
| Software, algorithm | *lmer4* | Bates et al. | | https://cran.r-project.org/web/packages/lme4/index.html |
| Software, algorithm | FlowJo v10.2 | Tree Star Inc., Ashland OR | RRID:SCR_008520 | |
| Other | | | | |

## Mice and antibiotic exposure

NOD/ShiLtJ mice (6 weeks old) were purchased from Jackson Laboratory (Bar Harbor ME), and bred in an SPF vivarium at the New York University Langone Medical Center (NYUMC) Skirball animal facility. All animal procedures were approved by the NYUMC Institutional Animal Care and Use Committee (IACUC protocol no. 160623). The dams and their litters were randomly assigned to control or PAT groups. At postnatal (P) day 23, the pups were weaned and housed to separate males and females. All mice received acidified drinking water supplied by the facility routinely except for the periods when some litters were receiving antibiotic treatment. A therapeutic dose of the macrolide tylosin tartrate (Sigma-Aldrich, Billerica MA) was given to mice in their non-acidified drinking

water with 333 mg/L (about 50 mg/kg body weight/day) (*Livanos et al., 2016*) on P5-10 for 1PAT mice or in three courses (P10-15, P28-31, and P37-40) (3PAT), exactly as described (*Livanos et al., 2016*).

## Diabetes monitoring

All mice were monitored for diabetes by weekly measurement of tail blood glucose using the Free-Style Lite meter and blood glucose test strips (Abbott Diabetes Care Inc., Abbott Park IL). The measurement was started at week 11 of age and continued to week 30; diabetes onset was defined as two consecutive values > 250 mg/dl (*Livanos et al., 2016*; *Wen et al., 2008*). Kaplan-Meier analysis was applied for evaluating diabetes progression of treatments (*Kaplan and Meier, 1958*), and the Log-rank (Mantel-Cox) test was applied to detect the difference significance between treatment (*Harrington and Fleming, 1982*).

## Collection of fecal and tissue samples

Individual mice were placed in an empty clean beaker for 2–5 min to allow them to defecate normally to obtain 3–4 pellets, which were frozen at –80°C for further analysis. At mouse sacrifice, the distal ileum (1 cm long) was collected, ileal contents removed, and tissue and contents separately introduced into RNA*later* (Qiagen, Valencia CA). The next most distal 1 cm ileal segment with contents was frozen at –80°C for 16S rRNA analysis, and the subsequent segment without contents was frozen at –80°C for histone modification analysis. Cecum samples with contents were frozen at –80°C for 16S rRNA and metabolic analyses. After removal of the colonic contents, colonic tissues were collected into RNA*later*, and the more proximal tissues and contents frozen at –80°C for microbiome 16S rRNA analysis. The liver of each mouse was collected and frozen at –80°C for metabolic analysis and histone modification analysis. From mice sacrificed at P42 and P70, the pancreas was removed and fixed in freshly prepared modified Bouin's fixative (*Leiter, 2001*), paraffin-embedded, sectioned, stained, and scored, as described (*Livanos et al., 2016*) with modification by using methyl green as counterstain (*Forestier et al., 2007*).

## Microbiome assessment with 16S rRNA

Fecal and intestinal microbiota DNA was extracted using the PowerLyzer PowerSoil DNA Isolation Kit (MoBio, Carlsbad CA) and the PowerSoil-htp 96 Well Soil DNA Isolation Kit (MoBio). The amplicon library of V4 regions of the bacterial 16S rRNA genes were obtained by triplicate PCR with barcoded fusion primers, quantification with the Qubbit 2.0 Fluorometer (Life Technologies, Carlsbad, CA), and combination of each DNA sample at equal concentrations as previously described (*Livanos et al., 2016*). The library was sequenced with the Ilumina MiSeq 2 × 150 bp paired end platform (Ilumina, San Diego CA) at the NYUMC Genome Technology Center.

## Microbiome community analysis

QIIME 2.0 was used as the amplicon read processing pipeline as described (https://qiime2.org/) (*Caporaso et al., 2010*). Reads with more than three consecutive low-quality bases (Phred score < 20) were filtered, and only reads with > 75% of the original length were retained. OTUs were picked using the open reference picking strategy based on the Greengenes database (*DeSantis et al., 2006*). Reads were first clustered into 97% identity OTUs using UCLUST program (*Edgar, 2010*), and taxonomy assignment was performed using the RDP Classifier with a confidence interval of 50%, and chimeras were removed using ChimeraSlayer (*Haas et al., 2011*). Microbial diversities within samples (α-diversity) and between samples (β-diversity), and taxa relative abundance were analyzed using QIIME2.0. α-diversity was evaluated with phylogenetic diversity (*Faith, 1992*), and mean values and statistical significance tests were calculated using Prism (GraphPad Software, La Jolla CA). β-diversity was evaluated with unweighted UniFrac (*Lozupone and Knight, 2005*). Statistical significance of the inter- and intra-group β-diversity was determined by permutation testing using R. Assessment of significantly different taxa between different treatment groups was performed using the ANCOM program in QIIME2.0 (*Mandal et al., 2015*). Mixed effects models (*Laird and Ware, 1982*) were fitted to test the disparity in relative abundance for each taxon (from phylum to genus) between two groups of mice (1PAT vs. 1PAT control and 3PAT vs. 3PAT control, respectively) for both males and females. The mixed effects models included relative abundances for each taxon

as outcomes, and group indicator and time as fixed effects, and the intercept and slope of the linear time trend for each mouse as random effects. Unclassified taxa and taxa which were monotonic or singletons in any of the groups (1PAT, 1PAT control, 3PAT and 3PAT control, for males and females, separately), were excluded in the analysis. For multiple testing correction, the Benjamini-Hochberg (BH) procedure (*Benjamini and Hochberg, 1995*) was applied for each taxonomic level.

## Microbiome assessment with whole genome shotgun sequencing

A total of 48 fecal samples from 24 mice (6 males and 6 females each from wither the 1PAT and 1PAT control groups at P12 and P49 were chosen for metagenomic study, along with each inoculum. Extracted genomic DNA (5 ng) from each sample was used for library preparation and subsequent whole genome sequencing (WGS) using the Illumina HiSeq 2500 platform. Samples were sequenced over 6 flow cell lanes as 100 bp paired-end reads. The metagenomes were pre-processed for quality metrics using *Trimmomatic* (*Bolger et al., 2014*) and aligned to the mouse genome (mm10) to reduce host-contaminated sequences using *KneadData* as previously described (*Schulfer et al., 2018*). After filtering low-quality and contaminated sequences, we performed functional profiling to detect microbial genes and pathways using *HUMAnN2 v0.9.5* (*Abubucker et al., 2012*) with default settings and screened against the EC-filtered UniRef90 database.

Raw shotgun reads were quality controlled using SHI7 (*Al-Ghalith et al., 2018*) and aligned using exhaustive gapped alignment at 95% identity (*Al-Ghalith and Knights, 2017*; *Needleman and Wunsch, 1970*) against a reference database of 21,186 putative BGCs predicted by antiSMASH or deposited in the MIBiG database (*Blin et al., 2017*; *Medema et al., 2015*; *Weber et al., 2015*). The per-sample metagenomic coverage of each BGC was calculated using in-house Python and R code and filtered to pathways with a ratio of actual coverage to expected coverage (expected coverage probability is defined as $1 - \exp(NL_{read}/L_{BGC})$, where $N$ = number of reads, $L_{read}$ = median read length, and $L_{BGC}$ = BGC sequence length) of at least 0.75. Differentiating BGCs were identified by comparing BGC presence/absence frequency between the treatment groups using Fisher's exact test with FDR correction at q < 0.15. To collapse homologous BGCs we used custom Python and C code to hierarchically cluster the pathways based on amino acid identity and open reading frame composition (*Rashidi et al., 2018*; *Shields-Cutler et al., 2018a*, *Shields-Cutler et al., 2018b*). Cluster annotations and taxonomic assignments were derived from their antiSMASH references. The metabolite pathways were discovered directly from the metagenomic data presented in the study, by analyzing coverage of the DNA pathways. The metagenomic data were annotated to particular pathways and taxa by DNA sequence homology ≥95% to BGC pathways present in the antiSMASH database (*Blin et al., 2017*), including Enterobacteriaceae reference strains.

## High resolution GC-MS analysis of cecal short chain fatty acids (SCFA)

Cecal contents (~10 mg) were subjected to a targeted GC-MS analysis to quantify short chain fatty acid levels, as described (*Lucas et al., 2018*; *Tangerman and Nagengast, 1996*), with modifications. Metabolite extraction was carried out by addition of 50:1 extraction solvent [80% methanol in water (LCMS Grade) with 0.5 mm zirconium/silica beads (Research Products International)] to the measured sample mass (±0.01 mg) in a tared bead blaster tube. Each vial underwent two homogenization cycles at 30 s on/30 s off at 6.0 m/s (4°C) and insoluble matter was pelleted by centrifugation at 21,000 g for 3 min at 4°C. The supernatant was transferred to a gas-tight 1.5 mL glass GC vial (Agilent Technologies) for analysis. Using aThermoTM TRACE 1310 gas chromatograph, a split injection method used a split ratio of 25, 1 μL injection volume, and split flow of 25.0 mL/min, with constant 250°C temperature for the injector with a Topaz low pressure drop precision liner with wool (RestekTM). Helium carrier gas was used at a constant flow of 1.0 mL/min with a ThermoTM TG-WAXMS A column (30m × 0.25 mm). The 15 min thermal gradient profile included equilibration of 2 min followed by a 1 min hold at 100°C, a ramp from 100 to 145°C at 20°C /min, a 4 min hold at 145°C, a ramp from 145 to 165°C at 15°C /min, finishing with a 3 min hold at 165°C. The GC system was coupled to a Thermo Q ExactiveTM mass spectrometer operating in electron ionization positive mode at 70KeV. MS1 scan range from 32 to 350 m/z was used at resolution 120,000 with AGC target 1e6 and maximum IT 100 ms. SCFA intensities were quantified at 5 ppm tolerance within a 0.1 min retention time window, using a 5-point standard curve (from 10 to 1000 pg/μL), with a randomized

acquisition order of samples and standards, run in duplicate. Standard curve analyte intensities were fit to a linear regression and sample values reported as picogram analyte/mg cecal material (ppm).

## GC-TOF-MS metabolomics of liver and serum

Aliquots of approximately 100 mg of liver tissue were placed in a 2 mL Magna Lyser tube (Roche) and mixed with 500 µL of ice-cold 50% acetonitrile, pulse sample 2 × 30 s @ 2000 in Magna Lyser (Roche). The homogenized mixture was centrifuged at 18,600 g for 5 min at 4°C to yield supernatant. The quality control samples for liver and serum were made by pooling individual liver supernatant into one tube and pooling individual serum sample into another tube. Aliquots of 100 µL liver supernatant or 30 µL serum sample were mixed with 1000 µL of a cold degassed acetonitrile-isopropanol-water solution (3:3:2), and centrifuged for 4 min at 18,600 g, and then dried by vacuum in new tubes. Study samples and quality control samples were processed identically. Study samples were randomized, and quality control pool samples (prepared under identical conditions) were interspersed. Samples were derivatized using a two-step method with acquisition parameters similar to Fiehn et al. (*Fiehn et al., 2008*; *Kind et al., 2009*). To summarize, a 0.5 µl volume was injected into an Agilent 6890 gas chromatograph (Agilent Technologies) with a 30 m long, 0.25 mm i.d. Rxi5Sil-MS column with 0.25 µm film thickness (Restek, Bellefonte PA), using a 250°C injector temperature in splitless mode with 25 s splitless time, at a constant flow of 1 ml/min. The oven temperature was ramped (20°C/min ramp) from 50°C to 330°C (*Chou et al., 2017*). Data were acquired using a Leco Pegasus 4D TOF-MS (Leco, Saint Joseph MI) with a 280°C transfer line temperature, electron ionization at –70 V, and an ion source temperature of 250°C. Spectra were acquired from m/z 50–750 at 20 spectra s-1 and 1850 V detector voltage.

GC-TOF-MS data were deconvoluted by ChromatTOF (Leco, St. Joseph MI) and then further processed by BinBase (*Skogerson et al., 2011*) for peak retention index calculations, spectral identification, and generation of a table of peak identifications and intensities. Multivariate data analysis was conducted using SIMCA 13.0 (Umetrics, Sweden) for data normalized to the sum of intensities. Mean-centered and pareto-scaled data were analyzed by Principal Component Analysis (PCA) and Orthogonal Projections to Latent Structures Discriminant Analysis (OPLS-DA). Peaks with Variable Influence on Projections (VIP) $\geq$ 1.0 were deemed important for differentiating the study groups. Significant changes in pairwise comparison were evaluated by Wilcoxon Rank-Sum test by SAS 9.4 (SAS Institute Inc., Cary NC) with p value $\leq$ 0.05 denoting statistical significance.

## RNA extraction and RNA-Seq

Total RNA was extracted from mouse tissues using the PureLink RNA Mini Kit (Invitrogen, Carlsbad CA), and contaminating genomic DNA was removed by treatment with DNase I (Qiagen). Total RNA quality and quantity were determined using the NanoDrop ND-1000 UV-Vis Spectrophotometer (NanoDrop Technologies, Inc., Wilmington DE), and Agilent 2100 Bioanalyzer (Agilent Technologies, Santa Clara CA). For RNA-Seq, we used methods, as described (*Ruiz et al., 2017*). Reads were aligned to the mouse GENCODE GRCm38.p5 (M14 release) genome using STAR v2.5.2b (*Dobin et al., 2013*), and the read summarization program FeatureCounts (*Liao et al., 2014*) was used to count mapped reads against annotated genes. Differential expression analysis between different treatments and KEGG pathways visualization was performed using DESeq2 in the R-package (*Love et al., 2014*) and Pathview, respectively (*Luo and Brouwer, 2013*). p values were corrected for multiple comparisons, based on the false discovery rate (FDR) (*Benjamini and Hochberg, 1995*) with significance considered by the adjusted p value<0.05. Differentially expressed pathways and functions were interpreted using with Ingenuity Pathway Analysis (IPA, QIAGEN Redwood City, http://www.ingenuity.com).

## Immune gene NanoString analysis

Immune gene expression profile of each of the above RNA samples was evaluated by using the nCounter GX mouse immunology kit (NanoString Technologies, Seattle WA). Counts were normalized using DESeq2 (*Love et al., 2014*). P values were corrected for multiple comparisons, based on the false discovery rate (FDR) (*Benjamini and Hochberg, 1995*), with significance considered by the adjusted p value<0.05. Heat maps were generated using the pheatmap package in R (*Kolde and Vilo, 2015*).

## RT-qPCR for host target gene expression

To evaluate specific gene expression, cDNA was synthesized through reverse transcription from 1 µg of each total RNA sample above with the Verso cDNA Synthesis kit (Thermo Scientific, Waltham MA) using random hexamer primers provided. qPCR was run in a LightCycler 480 system (Roche, Branchburg NJ) using 10 ng cDNA, target gene-specific primer pairs (*Supplementary file 7*) and Power SYBR Green PCR Master mix (Roche). Target mRNA was normalized to 18S rRNA or house-keeping gene HPRT as an internal control in each sample (*Saha and Blumwald, 2014*). For group mean comparisons, the Mann Whitney test was performed with p value < 0.05 as significant difference.

## Measurement of fecal IgA

Fecal samples were resuspended in PBS at a concentration of 50 mg/mL by extensive vortexing, allowed to stand for 20 min, and centrifuged at 16,000 *g* for 10 min to collect supernatant as described (*Haneberg et al., 1994*). Each supernatant was assessed for IgA using the mouse IgA ELISA kit with suitable dilutions, according to the manufacturer's instructions (Bethyl, Montgomery TX), and the absorbance was measured at a wavelength of 450 nm using the Dynex MRX TC Revela-tion microplate reader (Dynex Technologies, Chantilly VA) (*Ruiz et al., 2017*).

## Isolation of pancreatic and splenic leukocytes

The pancreas or splenic tissue of each mouse was placed into 5 mL of PBS supplemented with 2% fetal calf serum (Corning, Tewksbury MA), 1 mg/mL trypsin inhibitor (Sigma, St. Louis MO), 1 mg/mL of Collagenase IV (Worthington Biochemical, Lakewood, NJ), and 0.5 mg/mL DNaseI (Sigma) and the spleen in DMEM (Corning) supplemented with 10% fetal calf serum (FCS). The pancreas or spleen was physically disrupted for 5 min and dissociated using a GentleMACS Dissociator (Miltenyi Biotec Inc., Auburn CA). Following dissociation, the tissues were enzymatically digested for 20 min at 37°C. Following enzymatic digestion, the cell suspension was filtered, washed, and resuspended in cold PBS with 2% FCS. Spleens were physically disrupted over 70-micron nylon mesh (Corning) on ice. Cells were pelleted at 300 *g* for 5 min, supernatants were removed, followed by red blood cell lysis with ACK (ammonium-chloride-potassium) lysis buffer (Lonza, Basel, Switzerland). Single cell suspensions were washed and resuspended 1 mL of cold PBS with 1% FCS. Cells were stained using anti- CD45, CD4, CD8, CD3, CD44, CD62L, CD19 and B220 (BioLegend, San Diego CA). Data were acquired on a BD LSRII (BD Bioscience, San Jose CA) and analyzed using FlowJo v10.2 (Tree Star Inc., Ashland OR).

## Global histone modification analysis

Histones were isolated from flash-frozen whole post-mortem ileum and liver from P23 mice (n = 4 or 5 per sex and treatment group) and prepared for analysis by liquid chromatography coupled to tan-dem mass spectrometry (LC-MS/MS). Histone extraction, label-free chemical derivatization, and data acquisition using a Dionex Ultimate3000 nanoflow HPLC with a Waters nanoAcquity UPLC C18 col-umn (100 µm × 150 mm, 3 µm) online with aThermo Q-Exactive mass spectrometer using a data-independent acquisition method, as previously described (*Krautkramer et al., 2016*). Following data acquisition, normalization and quantification of histone PTM abundance was performed as pre-viously described (*Krautkramer et al., 2015*). Isobaric and co-eluting peptides were not deconvo-luted, and are denoted as such (e.g. K18ac+K23un and K18un+K23ac are isobaric and co-eluting and are denoted as a single value for K18ac/K23ac since their MS1 ions are identical and thus repre-sentative of both peptide species). Normalized percent of total values were then used to calculate fold-changes and statistics. All p values were generated using Welch's t test, with statistical signifi-cance set at p<0.05.

## Statistical associations between microbiota and host gene expression

Using the compPLS framework (*Ramanan et al., 2016*), we aimed to detect associations between taxa in the gut microbiome and host immune genes. To prevent detection of spurious associations, we: (i) performed a centered log-ratio (clr) transformation on the OTU relative abundance data; (ii) applied a variance decomposition to extract within-subject variation; and (iii) estimated a sparse lin-ear model via sparse Partial Least Squares (sPLS) regression to detect associations between a sparse

set of multi-collinear features (OTUs) and responses (host covariates) (*Bastien et al., 2005*; *Chun, 2010*). As our host covariates or response variables, we used expression levels representing significantly differentially expressed genes. We first filtered OTUs at the genus level with relative abundance > 0.01%. Taxa were then selected if present in at least one of the intestinal samples. This filtering resulted in a species-level OTU table. We decomposed the clr-transformed OTUs and host response data using a two-factor variance decomposition to account for differences in sex (M or F) and treatment group (1PAT and 1PAT control). For each sPLS run, we set the number of latent components to the number of non-zero singular values in the cross-covariance matrix. To find a sparse set of significant associations between OTUs and genes, we (i) applied sPLS and used a stability approach to regularization selection (StARS) (*Liu et al., 2010*) to select the sparsity level; and (ii) used bootstrap-based empirical p value calculation to assess the significance of associations of the StARS-selected support (*Lê Cao et al., 2009*). We calculated empirical p values over 5000 bootstraps and set a p value threshold of 0.05 after FDR multiple testing correction. We visualized significant associations as a network using the igraph package in R (*Csárdi and Nepusz, 2006*).

## Data deposition

RNA-Seq data that support the findings of this study have been deposited in ArrayExpress database (www.ebi.ac.uk/arrayexpress) with the accession code E-MTAB-6826 (https://www.ebi.ac.uk/arrayexpress/experiments/E-MTAB-6826). 16S rRNA data have been deposited in QIITA (https://qiita.ucsd.edu/) with the identifier 11242 (https://qiita.ucsd.edu/study/description/11242). Ileal NanoString data have been deposited in NCBI's Gene Expression Omnibus (https://www.ncbi.nlm.nih.gov/geo/) and are accessible through GEO Series accession number, GSE101721 (https://www.ncbi.nlm.nih.gov/geo/query/acc.cgi?acc=GSE101721). Shotgun metagenomics data have been deposited in the European Nucleotide Archive (ENA) (https://www.ebi.ac.uk/metagenomics/) under the accession number, PRJEB26585 (http://www.ebi.ac.uk/ena/data/view/PRJEB26585). Metabolomics data have been deposited at the NIH Common Fund Metabolomics Workbench (www.metabolomicsworkbench.org; doi: 10.21228/M8C39R).

## Abbreviations

Type 1 diabetes (T1D), pulsed therapeutic antibiotic treatment (PAT), non-obese diabetic mouse (NOD mouse), histone post translational modification (histone PTM).

## Acknowledgements

For assistance with these studies, we thank Savannah Costa, Yunzhu Li, Menghan Liu, Xuhui Zheng, James Zhou, Chad Trent, Matthew Farkouh, and Zhan Gao of New York University Medical Center, Shijia Zhu and Gang Fang of Mount Sinai Medical Center, NYU Langone Health Genome Technology Center, and NYU Metabolomics Core Resource Laboratory. These studies were supported by Janssen Labs London (15-A0-00-00-0039-29-01), NIH Grants [5T35DK007421, R37GM059785, F30 DK108494, T32GM008692, R01DK110014], −33.17CVD01 (TransAtlantic Networks of Excellence Program) from the Fondation Leducq, and the C and D fund.

## Additional information

### Funding

| Funder | Grant reference number | Author |
|---|---|---|
| National Institutes of Health | F30DK108494 | Kimberly A Krautkramer |
| National Institutes of Health | 5T35DK007421 | Sandy Ng<br>Rachel A Sibley |
| National Institutes of Health | R37GM059785 | John M Denu |
| National Institutes of Health | R01DK110014 | Huilin Li |
| Janssen Research and Development | 15-A0-00-00-0039-29-01 | Martin J Blaser |

| Fondation Leducq | -33.17CVD01 | Martin J Blaser |
|---|---|---|
| The C & D fund | | Martin J Blaser |

The funders had no role in study design, data collection and interpretation, or the decision to submit the work for publication.

## Author contributions

Xue-Song Zhang, Conceptualization, Formal analysis, Supervision, Validation, Investigation, Visualization, Writing—original draft, Project administration, Writing—review and editing; Jackie Li, Sandy Ng, Rachel A Sibley, Shawn Jindal, Formal analysis, Investigation, Visualization; Kimberly A Krautkramer, Formal analysis, Visualization, Methodology, Writing—original draft, Writing—review and editing; Michelle Badri, Kelly V Ruggles, Software, Formal analysis, Visualization, Methodology, Writing—review and editing; Thomas Battaglia, Data curation, Software, Formal analysis, Visualization, Methodology, Project administration, Writing—review and editing; Timothy C Borbet, Wimal Pathmasiri, Formal analysis, Visualization, Writing—review and editing; Hyunwook Koh, Software, Formal analysis, Methodology, Writing—review and editing; Yuanyuan Li, Data curation, Formal analysis, Investigation, Visualization; Robin R Shields-Cutler, Data curation, Software, Formal analysis, Visualization, Methodology, Writing—original draft, Writing—review and editing; Ben Hillmann, Gabriel A Al-Ghalith, Formal analysis, Visualization, Methodology; Victoria E Ruiz, Formal analysis, Investigation; Alexandra Livanos, Investigation, Writing—review and editing; Angélique B van 't Wout, Nabeetha Nagalingam, Formal analysis, Investigation, Writing—review and editing; Arlin B Rogers, Formal analysis, Investigation, Methodology; Susan Jenkins Sumner, Formal analysis, Supervision, Funding acquisition, Methodology, Writing—review and editing; Dan Knights, John M Denu, Richard Bonneau, Formal analysis, Supervision, Methodology, Writing—review and editing; Huilin Li, Formal analysis, Supervision, Investigation, Writing—review and editing; R Anthony Williamson, Marcus Rauch, Supervision, Funding acquisition, Investigation, Project administration, Writing—review and editing; Martin J Blaser, Conceptualization, Formal analysis, Supervision, Funding acquisition, Validation, Investigation, Visualization, Methodology, Writing—original draft, Project administration, Writing—review and editing

## Author ORCIDs

Xue-Song Zhang (iD) https://orcid.org/0000-0001-5080-0098
Jackie Li (iD) http://orcid.org/0000-0002-1376-6692
Martin J Blaser (iD) http://orcid.org/0000-0003-2447-2443

## Ethics

Animal experimentation: This study was performed in strict accordance with the recommendations in the Guide for the Care and Use of Laboratory Animals of the National Institutes of Health. All of the animals were handled according to approved institutional animal care and use committee (IACUC) protocols (160623) of the New York University Langone Medical Center.

## Decision letter and Author response

Decision letter https://doi.org/10.7554/eLife.37816.058
Author response https://doi.org/10.7554/eLife.37816.059

# Additional files

## Supplementary files

• Supplementary file 1. Microbial metabolic pathways significantly differentiated by 1PAT based on metagenomic analysis.
DOI: https://doi.org/10.7554/eLife.37816.038

• Supplementary file 2. List of metabolites differentially regulated by 1PAT in serum and liver early in life.
DOI: https://doi.org/10.7554/eLife.37816.039

• Supplementary file 3. Lists of ileal genes differentially expressed by 1PAT at P23 in males and females, based on RNA-Seq analysis.
DOI: https://doi.org/10.7554/eLife.37816.040

• Supplementary file 4. Lists of ileal immune genes differentially expressed from P12-P23 and P23-P42 in control and 1PAT males.
DOI: https://doi.org/10.7554/eLife.37816.041

• Supplementary file 5. Data of hepatic and ileal histone PTM state analysis.
DOI: https://doi.org/10.7554/eLife.37816.042

• Supplementary file 6. Ileal immune genes significantly differentially expressed between 1PAT and control in P23 male and female NOD mice.
DOI: https://doi.org/10.7554/eLife.37816.043

• Supplementary file 7. List of primer sequences used for RT-qPCR analysis.
DOI: https://doi.org/10.7554/eLife.37816.044

• Transparent reporting form
DOI: https://doi.org/10.7554/eLife.37816.045

### Data availability

RNA-Seq data that support the findings of this study have been deposited in ArrayExpress database (www.ebi.ac.uk/arrayexpress) with the accession code E-MTAB-6826 (www.ebi.ac.uk/arrayexpress/experiments/E-MTAB-6826). 16S rRNA data has been deposited in QIITA (https://qiita.ucsd.edu/) with the identifier 11242 (https://qiita.ucsd.edu/study/description/11242). Ileal NanoString data have been deposited in NCBI's Gene Expression Omnibus (www.ncbi.nlm.nih.gov/geo/) and are accessible through GEO Series accession number, GSE101721 (www.ncbi.nlm.nih.gov/geo/query/acc.cgi?acc=GSE10171). Shotgun metagenomics data have been deposited in the European Nucleotide Archive (ENA) (www.ebi.ac.uk/metagenomics/) under the accession number, PRJEB26585 (www.ebi.ac.uk/ena/data/view/PRJEB26585). Metabolomics data have been deposited at the NIH Common Fund Metabolomics Workbench (www.metabolomicsworkbench.org; doi: 10.21228/M8C39R)

The following datasets were generated:

| Author(s) | Year | Dataset title | Dataset URL | Database, license, and accessibility information |
|---|---|---|---|---|
| Thomas Battaglia, Xue-Song Zhang, Martin J Blaser | 2018 | Ileal transcriptome profiles altered by early-life single pulse antibiotic exposure | www.ebi.ac.uk/arrayexpress/experiments/E-MTAB-6826 | Publicly available at the Electron Microscopy Data Bank (accession no: E-MTAB-6826) |
| Thomas Battaglia, Xue-Song Zhang, Martin J Blaser | 2018 | Intestinal microbiome altered by early-life pulse antibiotic exposure | https://qiita.ucsd.edu/study/description/11242 | Publicly available at Qiita (identifier: 11242) |
| Thomas Battaglia, Xue-Song Zhang, Martin J Blaser | 2018 | Intestinal immunity pathways altered by early-life single pulse antibiotic exposure | www.ncbi.nlm.nih.gov/geo/query/acc.cgi?acc=GSE101721 | Publicly available at the NCBI Gene Expression Omnibus (accession no: GSE101721) |
| Thomas Battaglia, Xue-Song Zhang, Martin J Blaser | 2018 | Intestinal metagenome and metabolic pathways altered by early-life single pulse antibiotic exposure | www.ebi.ac.uk/ena/data/view/PRJEB26585 | Publicly available at the European Nucleotide Archive (accession no: PRJEB26585) |
| Yuanyuan Li, Xue-Song Zhang, Susan Jenkins Sumner, Martin J Blaser | 2018 | Metabolomics involved in early-life single pulse antibiotic exposure | www.metabolomicsworkbench.org/data/DRCCMetadata.php?Mode=Project&ProjectID=PR000660 | Publicly available at Metabolomics Workbench (10.21228/M8C39R) |

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
