## [Decision Letter]

Thank you for submitting your article "Antibiotic-induced acceleration of type 1 diabetes alters maturation of innate intestinal immunity" for consideration by *eLife*. Your article has been reviewed by two peer reviewers, and the evaluation has been overseen by a Reviewing Editor and Wendy Garrett as the Senior Editor. The following individual involved in review of your submission has agreed to reveal his identity: Gerard Eberl (Reviewer #2).

The reviewers have discussed the reviews with one another and the Reviewing Editor has drafted this decision to help you prepare a revised submission.

Summary:

The reviewers agreed that this paper represents an interesting study of how early-life antibiotic exposure can impact type 1 diabetes incidence in later life. The authors show that treatment of mice with macrolide antibiotics between 5 and 10 days after birth lead to an acceleration of the development of type 1 diabetes in male mice. They employ several "omics" approaches to investigate the effects of antibiotic treatment on gut microbiota composition and function, intestinal gene expression, host metabolism and immune cell frequencies. Their findings reveal that pathogenesis is associated with a persistent alteration of the microbiota, the emergence of biosynthetic gene clusters (BGC), a decrease in short chain fatty acid production by the microbiota, an alteration of the serum and liver metabolome, an increase in immunity-related pathways in the intestinal transcriptome, in particular in genes coding for innate immune pathways. These immune alterations lead to a decrease in FoxP3 and IgA expression in the intestine and an increase in B cells in the pancreas. Their findings underscore the complexity of the microbiota-host interaction and demonstrate that antibiotic perturbation of this crosstalk early in life can have lasting consequences for the host.

The reviewers felt that overall, this is an extremely informative piece of work. Although the authors used a multi-'omics approach to generate hypotheses rather than to evaluate hypotheses about specific mechanisms, the reviewers felt that the work provides valuable insight to the gut microbiome field. However, there are some aspects of the data analysis and results interpretation that could be improved.

Essential revisions:

1) Subsection “PAT directionally alters the metagenome and its metabolic products”, second paragraph and Figure 2E: The authors have not provided a strong Rationale for why they chose to focus on Biosynthetic Gene Clusters (BGCs) to investigate predicted differences in metabolic activity of the microbiota. This approach only allows for prediction of secondary metabolites, and does not consider other metabolic pathways that might have impact on host biology for example synthesis of bacteria cell surface components. These predicted PAT associated effects on secondary metabolites are not discussed further in the rest of the manuscript so their importance in 1PAT effects on autoimmunity in the NOD model is not clear. Is there any evidence that arylpolyenes, siderophores, and NRSPs produced by Enterobacteria are immune modulators or are associated with autoimmunity?

2) "[…] predicting specific metabolites linked to the taxa that were independently identified in the analysis of 16S relative abundances". This sentence is confusing and needs revision. All secondary metabolites identified as differentially abundant in the BGC study were assigned to *Enterobacteriaceae*. However, the 16S dataset identified many more taxa as differentially abundant under 1PAT conditions (Figure 2C). What is the evidence that these metabolites are associated with *Enterobacteriaceae*? Were these metabolites also assigned to these taxa in the metagenomics data presented in the study?

3) "These sex-specific differences, preceding any antibiotic exposure, provide an important opportunity in future studies to better understand the basis of the T1D gender dimorphism in NOD mice." (Also Discussion, fourth paragraph). This statement is incorrect in multiple ways and needs revision. a) The terms "sex" and "gender" have distinct meanings, and are not interchangeable. The relevant term in this study is "sex" as the evaluation is based on biological differences between male and female mice. Gender refers to social and cultural influences on behavior and physiology. b) Markle et al., Science. 2013 Mar 1;339(6123) showed that sex differences in microbiome composition can modify autoimmune diabetes in the NOD model and that these effects were testosterone-dependent. The highlighted statement is an over-interpretation of the observed male vs. female changes in ileal gene expression data measured at post-natal day 2 (before puberty).

4) Figure 6B (Foxp3 expression): The data show a striking decrease in Foxp3 expression level in day 15 ileal samples, in both Control and 1PAT samples. Is there prior evidence for this temporal pattern in Foxp3 expression? How do the Authors think this observation impacts autoimmunity in this mouse model?

5) Subsection “PAT-induced changes in adaptive immunity”: The authors do not provide a rationale for their decision to analyze T cells and B cells in the spleens but not in purified pancreatic islets. The T cell dependence of this disease model is well established. Did the Authors examine the effects of 1PAT on frequency and function of Foxp3+ Treg or Th17 cells? How do the authors reconcile the increase in splenic CD62L^+^ T cells (and reported decrease in activated CD44+ T cells) in the 1PAT vs Control with T1D development? Were CD4^+^ T cells and CD8^+^ T cell frequencies and absolute numbers changed by 1PAT in lymphoid organs and most critically, in the pancreas? As all cell frequencies are relative, these data should also be reported as absolute cell numbers.

6) "[…] the mLNs and spleen, and in the target organ (pancreas), preceding T1D development" – there are no immunophenotyping data provided for the mLN. Please add these data.

7) "taxa that are highly metabolically active". There is no direct evidence that changes in metabolites are associated with differentially abundant taxa. This statement needs to be changed to reflect the actual data presented.

8) "This study revealed a small consortium of pathobionts (*Enterococcus, Blautia*, Enterobacteriaceae species, and *Akkermansia)": Akkermansia* is not considered to be a pathobiont. Rather it is a microbe whose abundance is inversely correlated with obesity, inflammation and metabolic syndrome (Cani and de Vos, 2017).

9) "The more substantial effects of 1PAT on hepatic than ileal chromatin may reflect the cumulative effects of the altered microbiome on both metabolism as well as abnormal signaling from intestinal cells" – this sentence is confusing and should be removed. As the authors know ABX metabolism is in the liver, and likely a major contributor to the observed effects on liver gene expression and histone modifications.

10) We suggest changing the last sentence of the Abstract that mentions "mechanisms", as no mechanism is reported as such.

---

## [Author Response]

Essential revisions:1) Subsection “PAT directionally alters the metagenome and its metabolic products”, second paragraph and Figure 2E: The authors have not provided a strong Rationale for why they chose to focus on Biosynthetic Gene Clusters (BGCs) to investigate predicted differences in metabolic activity of the microbiota. This approach only allows for prediction of secondary metabolites, and does not consider other metabolic pathways that might have impact on host biology for example synthesis of bacteria cell surface components. These predicted PAT associated effects on secondary metabolites are not discussed further in the rest of the manuscript so their importance in 1PAT effects on autoimmunity in the NOD model is not clear. Is there any evidence that arylpolyenes, siderophores, and NRSPs produced by Enterobacteria are immune modulators or are associated with autoimmunity?

Thank you for these comments. We explored the BGCs in addition to the metabolic pathway profiling because they encode complex metabolites that possess a range of biological activities that mediate microbial communities and interactions with their environments. As the reviewers’ have suggested, we now extend our discussion about the BGC results by adding the following words into the Discussion section:

“Bacterial siderophore and related transition-metal scavenging responses are often associated with pathobiont organisms and activity, and have been shown to be important mediators of bacterial community structure and of bacterial persistence within human hosts during infection. […] NRPS family products are broad in structure but include metabolites with demonstrated anti-inflammatory and immunosuppressant activity such as cyclosporin A (Felnagle et al., 2008)”.

2) "[…] predicting specific metabolites linked to the taxa that were independently identified in the analysis of 16S relative abundances". This sentence is confusing and needs revision. All secondary metabolites identified as differentially abundant in the BGC study were assigned to Enterobacteriaceae. However, the 16S dataset identified many more taxa as differentially abundant under 1PAT conditions (Figure 2C). What is the evidence that these metabolites are associated with Enterobacteriaceae? Were these metabolites also assigned to these taxa in the metagenomics data presented in the study?

We thank the reviewer for drawing our attention to this sentence and we have reworded it as follows:

“In total, these studies indicate a directional (not substitutional) effect of PAT on the metabolite profiles as detected by metagenomic analyses, and are consistent with changes in taxa that were independently identified in the analysis of 16S relative abundances.”

We have also clarified the method that revealed these associations, as “The metabolite pathways were discovered directly from the metagenomic data presented in the study, by analyzing coverage of the DNA pathways. The metagenomic data were annotated to particular pathways and taxa by DNA sequence homology >95% to BGC pathways present in the antiSMASH database (Blin et al., 2017), including *Enterobacteriaceae* reference strains”. This is how the pathways we have identified are linked to *Enterobacteriaceae* (with the exception of the one *Enterococcus* product, subsection “PAT directionally alters the metagenome and its metabolic products”, fourth paragraph). It is possible that further pathways exist in enriched organisms from the 16S dataset, but the BGC pathway analysis was restricted to the metagenomic samples in order to observe the metabolite pathway coverage directly.

As such, we now have revised the first sentence in the BGC Results paragraph to read:

“Since secondary metabolites are bioactive small molecules affecting microbial community structure and/or host physiology (Dorrestein, 2014; Sharon et al., 2014), we then asked whether the metagenomic analysis could also identify biosynthetic gene clusters (BGCs) encoding significantly differential secondary metabolites. Using an accelerated optimal gapped alignment algorithm, we mapped the metagenomic reads against a BGC database and identified 228 BGCs with high rates of within-sample metagenomic coverage.”

3) "These sex-specific differences, preceding any antibiotic exposure, provide an important opportunity in future studies to better understand the basis of the T1D gender dimorphism in NOD mice." (Also Discussion, fourth paragraph). This statement is incorrect in multiple ways and needs revision. a) The terms "sex" and "gender" have distinct meanings, and are not interchangeable. The relevant term in this study is "sex" as the evaluation is based on biological differences between male and female mice. Gender refers to social and cultural influences on behavior and physiology.

Thank you for your correction; we agree with you. We now have corrected the text by using “…sex …” throughout.

b) Markle et al., Science. 2013 Mar 1;339(6123) showed that sex differences in microbiome composition can modify autoimmune diabetes in the NOD model and that these effects were testosterone-dependent. The highlighted statement is an over-interpretation of the observed male vs. female changes in ileal gene expression data measured at post-natal day 2 (before puberty).

We now provide a more cautious interpretation of the P2 findings by revising the text:

“These sex-specific differences at post-natal day 2, preceding any antibiotic exposure and puberty, may provide an important opportunity in future studies to better understand the basis of the T1D sex dimorphism in NOD mice before puberty.”

4) Figure 6B (Foxp3 expression): The data show a striking decrease in Foxp3 expression level in day 15 ileal samples, in both Control and 1PAT samples. Is there prior evidence for this temporal pattern in Foxp3 expression? How do the Authors think this observation impacts autoimmunity in this mouse model?

Thank you for pointing out this. Based on your comment, we checked the raw data, and realized that we made an error in reporting the Foxp3 expression in P15 and P42 mice. The calculations we used for each gene in question is to use the expression level in relation to the control at each timepoint. In each case, the control serves as the reference, defined as a value of 1.0. We mistakenly did not properly convert the values into the ratio with controls for P15 and P42. We have checked all of the other primary qPCR data, and this was the only error we found. We now show the correct data, in which not surprisingly, P15 looks similar in direction to P12; and we now include this to replace the panel in Figure 6B.

5) Subsection “PAT-induced changes in adaptive immunity”: The authors do not provide a rationale for their decision to analyze T cells and B cells in the spleens but not in purified pancreatic islets. The T cell dependence of this disease model is well established.

We did not directly purify the pancreatic islets during the sacrifices, since we used the pancreatic tissues for pancreatic islet histology analysis (which provides a visual quantitation of the local islet inflammation and tissue destruction; See Figure 1C), and for pancreatic T cell and B cell analyses. In future studies, we will attempt to purify pancreatic islets to examine the T cell populations that are present.

Did the Authors examine the effects of 1PAT on frequency and function of Foxp3+ Treg or Th17 cells?

Yes. We evaluated small intestinal LPL Treg and Th17 subsets at P42, which is 32 days after antibiotic exposure, and did not observe any significant difference between 1PAT and 1PAT control. Because of this long time window, we may have missed immunological changes that happened earlier. In future studies, we plan to examine the tissues during earlier time points to capture the dynamics of these changes.

How do the authors reconcile the increase in splenic CD62L^+^ T cells (and reported decrease in activated CD44+ T cells) in the 1PAT vs Control with T1D development?

The pancreas was analyzed as the site of T1D immunopathogenesis, while mLNs represent the lymph nodes draining the pancreas and the gut, and the spleen was used to gauge changes in systemic adaptive immunity that may have been altered as a result of the observed differences in innate immunity during early life with 1PAT. Evaluating the spleen, we found increased frequencies of CD62L^+^ CD4^+^ and CD62L^+^ CD8^+^ T cells (Figure 7—figure supplement 1B) cells in the 1PAT mice, but we do not see this in the draining lymph nodes nor in the pancreas, consistent with the accepted T cell dependent model of pancreatic immunopathogenesis (Pearson et al., 2016). Although we did not observe more activated T-cells, we observed more B-cells in the pancreas. The observed increase in frequency of naïve T cells in the spleen indicates systemic changes in adaptive immune development and this suggests decreased T cell differentiation into the separate activated T cell subsets. As we observed these changes in naïve T cell frequency in the spleen, but not in the gut draining lymph nodes or in the pancreas, it is likely that this systemic 1PAT phenotype is a result of the altered innate immune development we saw following 1PAT exposure treatment. Consistent with these findings, we observed decreased fecal IgA (as we found with 1PAT exposure in C57BL/6 mice (Ruiz et al., 2017), suggesting reduced T cell-dependent initiation of B cell class switching (Chorny et al., 2010). However, the pancreatic immune environment still maintained activated CD4 and CD8 T cells without significant changes in CD44 or CD62L expression. The increased B cells in the pancreatic tissue may be playing a role in immunopathogenesis, as B cells have been shown to participate in the generation of autoreactive T cells in NOD mice (Serreze et al., 1996).

Were CD4^+^ T cells and CD8^+^ T cell frequencies and absolute numbers changed by 1PAT in lymphoid organs and most critically, in the pancreas? As all cell frequencies are relative, these data should also be reported as absolute cell numbers.

When we examined absolute numbers of CD4 and CD8 T cells in all organs at P42, we observed no significant differences between control and 1PAT (Author response image 1). However, we observed altered frequencies, which indicate that there is a change in the sub-populations of the lymphocytes present (Figure 7—figure supplement 1B). As discussed above, it is possible that we missed the window for early events in immunopathogenesis, since P42 mice already are adults. The ileal gene expression studies (Figures 4, 5, and 6A, B) point to significant changes already occurring much earlier in life (P12, P15, and P23), which will be a focus of future experiments. In the current draft, we added “Differences in absolute count of lymphocytes were not observed in T cells or B cells, indicating there were no changes in lymphocyte quantity (data not shown). However, the change in frequency of naive and activated T cells suggests that the observed differences in the systemic differentiation of these cells does not impact total cellularity.”

6) "[…] the mLNs and spleen, and in the target organ (pancreas), preceding T1D development" – there are no immunophenotyping data provided for the mLN. Please add these data.

Upon re-review, we re-examined the mLN data and found that the differences were not significant, and that we exclusively observed significantly increased frequencies of naïve T cells exclusively in the spleen. We now have corrected the revised text by removing “the mLNs” and “In the mLNs, the PAT mice showed increased CD4^+^ T cells expressing (naïve) CD62L^+^ and CD44+ CD4^+^ T cells were decreased (data not shown).” from this section.

7) "taxa that are highly metabolically active". There is no direct evidence that changes in metabolites are associated with differentially abundant taxa. This statement needs to be changed to reflect the actual data presented.

Thank you.We are sorry that this was not clear. The rationale for our statement in the text is our observation about the directionality of the metabolomics data. As indicated in the text“Notably, of the 131 pathways differentiating 1PAT and control mice, 97% were overrepresented in the 1PAT samples, significantly deviated from chance at both times in both males and females (p < 0.001 for each subanalysis) ((Figure 2—figure supplements 8 and 9).” This significant asymmetry is the basis for our statements which we have now have revised to indicate:

“We now show in NOD mice that the gut microbiome was substantially remodeled by a single early-life PAT exposure, losing diversity without recovery over the entire window relevant to the development of auto-immunity, and with selection for taxa that may be highly metabolically

active.”

“In total, our findings support the hypothesis that by selectively diminishing particular (beneficial) taxa, PAT exposure permits emergence of a less diverse microbiota (Figure 2B), dominated most likely by highly metabolically active host-interactive taxa.”

And we removed “selecting for a highly metabolically active metagenome,” from the Abstract.

8) "This study revealed a small consortium of pathobionts (Enterococcus, Blautia, Enterobacteriaceae species, and Akkermansia)": Akkermansia is not considered to be a pathobiont. Rather it is a microbe whose abundance is inversely correlated with obesity, inflammation and metabolic syndrome (Cani and de Vos, 2017).

Thank you for pointing out this. We agree. We have revised the text by changing the sentence into “This study revealed a small consortium of pathobionts (*Enterococcus, Blautia*, and Enterobacteriaceae species)….." After this sentence, we added “*Akkermansia*, a taxon that has been inversely correlated with obesity, inflammation, and metabolic syndrome (Cani and de Vos, 2017), also is present in increased relative abundance in PAT-exposed male mice.”

9) "The more substantial effects of 1PAT on hepatic than ileal chromatin may reflect the cumulative effects of the altered microbiome on both metabolism as well as abnormal signaling from intestinal cells" – this sentence is confusing and should be removed. As the authors know ABX metabolism is in the liver, and likely a major contributor to the observed effects on liver gene expression and histone modifications.

Thank you for your suggestion.To avoid confusing readers, we have removed the sentence in question from the text as suggested. In fact, in the 1PAT experiment, the antibiotic exposure ended at P10. At P23, 13 days after antibiotic exposure, antibiotic should be long-since gone, since the half-lifeof tylosin is about 1 hour in small animals (Plumb, 2015)

(https://toxnet.nlm.nih.gov/cgi-bin/sis/search/a?dbs+hsdb:@term+@DOCNO+7022).

10) We suggest changing the last sentence of the Abstract that mentions "mechanisms", as no mechanism is reported as such.

We agree. We have changed “…. identifies mechanisms….” into “reveals multiple potential pathways to understand pathogenesis ….”.

Additional References:

Chorny, A., Puga, I., and Cerutti, A. (2010). Innate signaling networks in mucosal IgA class switching. Adv Immunol 107, 31-69.

Plumb, D. (2015). Veterinary Drug Handbook 8th ed. (Ames IA: Wiiley-Blackwell).

Serreze, D.V., Chapman, H.D., Varnum, D.S., Hanson, M.S., Reifsnyder, P.C., Richard, S.D., Fleming, S.A., Leiter, E.H., and Shultz, L.D. (1996). B lymphocytes are essential for the initiation of T cell-mediated autoimmune diabetes: analysis of a new "speed congenic" stock of NOD.Ig mu null mice. J Exp Med 184, 2049-2053.